# M3LEO: A Multi-Modal, Multi-Label Earth Observation Dataset Integrating Interferometric SAR and Multispectral Data

**Matt Allen**
University of Cambridge, UK
mja78@cam.ac.uk

**Francisco Dorr**
Independent, Argentina
fran.dorr@gmail.com

**Joseph A. Gallego-Mejia**
Drexel University, USA
jagallegom@unal.edu.co

**Laura Martínez-Ferrer**
Universitat de València, Spain
laura.martinez-ferrer@uv.es

**Anna Jungbluth**
European Space Agency, Climate Office, UK
anna.jungbluth@esa.int

**Freddie Kalaitzis**
University of Oxford, UK
freddie.kalaitzis@cs.ox.ac.uk

**Raúl Ramos-Pollán**
Universidad de Antioquia, Colombia
raul.ramos@udea.edu.co

## Abstract

Satellite-based remote sensing has revolutionised the way we address global challenges in a rapidly evolving world. Huge quantities of Earth Observation (EO) data are generated by satellite sensors daily, but processing these large datasets for use in ML pipelines is technically and computationally challenging. Specifically, different types of EO data are often hosted on a variety of platforms, with differing degrees of availability for Python preprocessing tools. In addition, spatial alignment across data sources and data tiling for easier handling can present significant technical hurdles for novice users. While some preprocessed Earth observation datasets exist, their content is often limited to optical or near-optical wavelength data, which is ineffective at night or in adverse weather conditions. Synthetic Aperture Radar (SAR), an active sensing technique based on microwave length radiation, offers a viable alternative. However, the application of machine learning to SAR has been limited due to a lack of ML-ready data and pipelines, particularly for the full diversity of SAR data, including polarimetry, coherence and interferometry. In this work, we introduce M3LEO, a multi-modal, multi-label Earth observation dataset that includes polarimetric, interferometric, and coherence SAR data derived from Sentinel-1, alongside multispectral Sentinel-2 imagery and a suite of auxiliary data describing terrain properties such as land use. M3LEO spans approximately 17M data chips, each measuring 4x4 km, across six diverse geographic regions. The dataset is complemented by a flexible PyTorch Lightning framework, with configuration management using Hydra, to accommodate its use across diverse ML applications in Earth observation. Additionally, we provide tools to process any dataset available on popular platforms such as Google Earth Engine for seamless integration with our framework. We show that the distribution shift in self-supervised embeddings is substantial across geographic regions, even when controlling for terrain properties. Data is available at `huggingface.co/M3LEO`, and code at `github.com/spaceml-org/M3LEO`.

# 1 Introduction

Satellite-based Earth observation data is fundamental in addressing global problems in a rapidly changing world, offering large-scale, high resolution, high frequency data for applications from tracking wildfires [1] and deforestation [2] to refugee settlement mapping [3] and war zone damage assessment [4, 5]. Information from these tasks is critical in crafting responses to man-made [6] and environmental crises [7], but is constrained by the use of optical (wavelengths from visible to near-infrared) sensing data. Such sensors are unable to operate in adverse weather or cloudy conditions [8], or at night, limiting their usefulness for time-critical tasks such as natural disaster management [7], environmental protection [9] or maritime surveillance [10]. Synthetic Aperture Radar (SAR) sensing presents an alternative that is able to overcome these limitations.

Unlike optical sensors, SAR instruments actively illuminate terrain using microwave pulses, ensuring visibility without the need for daylight. These long-wavelength pulses can penetrate cloud cover and other adverse atmospheric conditions such as dust, making SAR-based sensing a valuable alternative to optical sensors for robust day-night coverage. Additionally, microwave radiation can penetrate some solid objects such as small-scale vegetation or soils — allowing, for example, measurements of properties relating to soil moisture under vegetation [11] or the identification of archaeological features hidden below ground [12]. In addition to exploiting the wavelength and active illumination of SAR data, the complex nature of SAR signals — returning both amplitude and phase — can also be leveraged to provide insights beyond what is possible using optical data. Coherence, the complex correlation between pairs of SAR acquisitions, has been used successfully for tasks including flood detection [13, 14], detection of urban damage [15] and forest canopy height measurement [16]. The phase difference between co-registered SAR acquisitions, measured through interferometry, enables the detection of surface height changes with millimetre accuracy, independently of the horizontal resolution of the sensor. This capability is critical for monitoring geological phenomena such as earthquakes [17], landslides [18], glacial movement [19, 20], magma chamber growth [21] and infrastructure deformation [22].

While SAR offers opportunities to overcome the limitations of optical sensors, and to give insights that are impossible to provide using visible wavelengths, it is associated with substantial additional complexity. The automated analysis of optical data, sometimes used in fusion with SAR data, has seen great success in recent years [23–25] — including the development of large foundation models able to make use of planetary-scale datasets [26, 27]. The application of large-scale deep learning to SAR data without simultaneous use of optical data, however, is more limited [28]. The complexities of processing SAR data, particularly estimating coherence and performing interferometry — which require processing phase information as using complex numbers — mean that the full diversity of SAR data types is not available at scale in formats compatible with machine learning (ML) pipelines.

To address these challenges we introduce M3LEO, a large-scale multi-modal Earth Observation dataset comprising polarimetric, interferometric and coherence data for SAR as well as Sentinel-2 data and auxiliary datasets describing surface properties such as land use, biomass measurements and elevation. We provide this data pre-processed as ML-readable, tiled images to abstract complex parameter choices typically made by domain experts. We also include a flexible PyTorch Lightning framework, parameterised using Hydra [29], to further minimise barriers to usage. We include a preliminary analysis on distribution shift for terrain properties and the appearance of SAR data across geographic regions. Finally, in addition to the pre-formatted data we offer for download, we provide tools enabling ML practitioners to process any data retrievable from Google Earth Engine into the same tiled format, such that it can be used in our framework.

# 2 Related Work

Deep learning has been applied over the last decade to curated optical imagery with great success [30–33], including the recent development of large, self-supervised foundation models [34–37]. Such models have been extraordinarily successful in tasks such as semantic segmentation [38, 39], image classification [33], [38] [40] and object detection [38, 41]. EO data from optical sensors has similarly been the subject of success for deep learning practitioners. Early work focusing on small, fully supervised models, showed great promise in a huge range of tasks, including land cover classification [23], biomass measurement [24], road detection [25, 42] and flood mapping [43], although many models were limited in scope to a small geographic area [24, 44–46]. More recent work has focused

on the development of large foundation models, often self supervised [28, 47], which are readily adaptable to a range of downstream tasks and geographic areas [27].

Work applying these models to optical EO data has been enabled by the wealth of easily accessible open data. As well as being available as raw products from satellite data providers such as ESA [48], many datasets comprising optical satellite imagery in ML-ready formats exist [49–52], across a range of spatial resolutions [53, 54], and for multiple time-points [50, 55], although they may be limited in other scopes — for example, not having aligned task labels [55, 56] or being limited to a single region [50]. We provide data at a large scale (14.1% of the land surface of the Earth) with a diverse set of auxiliary labels.

The application of deep learning to SAR data is less comprehensive. A body of work applying deep learning directly to SAR data exists [57–60] [61], but data limitations mean that geographic or temporal generalisability, often lacking in remote sensing models [62], has not clearly been shown. The development of foundation models for SAR may prove to be productive in obtaining geographic and temporal generalisability. To create foundation models, SAR is commonly applied alongside optical imagery in data fusion-based approaches [63, 64], but with little attribution regarding whether such approaches can work well without optical data. Some work exists exploiting schemes such as masked autoencoding [65, 66], contrastive learning [67], or knowledge distillation [68, 69] to develop foundation models for polarimetric SAR — and shows that strong geographic generalisabilty is obtainable when using SAR data at large scales [65, 69]. Many datasets providing ML-readable polarimetric SAR (polSAR) data exist [51, 55, 70–75], although most do not provide interferometric SAR (inSAR) data [74, 75].

The full diversity of inSAR datatypes, such as interferometry and coherence, have seen a number of applications in machine learning, and a rich tapestry of applications in other contexts. Interferometry, for example, is often used to track earthquakes [17], landslides [18] and glacial movement [19, 20], in a manner that is both more repeatable — being immune to adverse weather conditions — and more accurate — being able to track millimetre-scale height changes — than methods using optical data. Coherence has been used with success in urban damage assessment [15], flood detection [13, 14] and canopy height measurement [16].

A number of datasets exist making these datatypes available to deep learning users, many of which are focused on specific events, tasks or locations. Hephaestus [76], for example, contains 216K interferometric SAR (inSAR) patches localised to volcanoes annotated with various labels describing volcanic activity. ISSLIDE [77] contains inSAR data from the French alps describing slow landslides, and Pol-InSAR-Island [78] comprises inSAR data describing land cover on Baltrum, a Frisian island. UrbanSARFloods [79] contains Sentinel-1 interferometric coherence data from a diverse set of global locations, but is limited to specific events. S1SLC_CVDL [80] opts to provide complex-valued single-look SAR data rather than processed interferometric data, from three manually selected Sentinel-1 scenes containing major population centres. We make inSAR and coherence data available at a multi-continental scale, alongside both polarimetric SAR and optical data, in M3LEO.

## 2.1   Comparison to Existing Datasets

We provide a comparison between a number of popular large-scale Earth observation datasets, including M3LEO, in Table 1. We define *tile* to mean a fixed location or area on the surface of the Earth, and *chip* as the content of some data product over that tile. Unlike in many of these datasets, we provide acquisitions from different seasons within the same year for the same tile as channels, rather than separate chips. The described number of chips in Table 1 therefore appears relatively lower for M3LEO compared to, for example, SSL4EO-S12, for a fixed number of satellite acquisitions. We instead measure the number of timepoints in years, rounded up for part-years. Of the four datasets offering Sentinel-1 SAR data, only M3LEO offers data from multiple years, and only M3LEO offers inSAR data (although see Section 1 for a brief description of available task-specific or localised interferometric datasets).

Regarding spatial coverage, of the datasets listed in Table 1 M3LEO is most similar in scale to SatlasPretrain and SSL4EO-L, with these three datasets being significantly larger than the remainder. Of these three datasets, only M3LEO provides SAR data of any modality.

The temporal coverage of M3LEO sits between SatlasPretrain and SSL4EO-L, although the aggregate number of years does not give a full picture — in SatlasPretrain, some acquisitions are available for

a wider range of years for specific events, and in SSL4EO-L different data products are sometimes collected for non-overlapping years, meaning much of the dataset is not a parallel corpus. In M3LEO, the primary satellite data from Sentinel-1 and Sentinel-2 is available for 2018-2020 for all tiles.

Several of these datasets contain auxiliary information in addition to satellite acquisitions. Land cover labels are common, included in SSL4EO-L, SEN12MS, BigEarthNet and SatlasPretrain [51, 56, 72, 73]. Additional auxiliary datasets are sometimes available — SSL4EO-L, for example, includes more detailed crop classification data. SatlasPretrain introduced a number of novel additional labelled datasets including building and road polygons. M3LEO currently includes 4 auxiliary datasets - Land cover, vegetation cover, aboveground biomass and Digital Elevation Models (DEMs).

Many datasets are pre-sampled within their selected AOIs prior to distribution. Some datasets sample based on a manually specified distribution (for example, SSL4EO-S12 and SSL4EO-L sample locations based on Gaussian distributions centred on large cities), and some randomly. We include all available tiles within our AOIs — partly with a view to increasing data volume, but also to enable further research on sampling schemes. Selecting a good sampling strategy to diversify actively illuminated radar data with phase information is not straightforward, and it doubtful whether sampling schemes developed for optical EO imagery would transfer well to this data. The auxiliary data included in M3LEO, such as elevation and land use, may be useful for constructing such sampling schemes.

Table 1: **Existing Datasets.** Summary of popular large-scale Earth observation pre-training datasets.

| Dataset | SAR | Years | Num. Tiles | Num. Chips[*] | Tile Size km | Coverage km$^2$ | Sampling |
|---------|-----|-------|------------|---------------|--------------|-----------------|----------|
| SSL4EO-S12 [55] | Y | 1 | 251K | 3M | 2.64×2.64 | $1.75\times10^6$ | Targeted |
| SSL4EO-L [51] | N | 6 | 250K | 5M | 7.92×7.92 | $1.57\times10^7$ | Targeted |
| SEN12MS [70] | Y | 1 | 181K | 542K | 2.56×2.56 | $1.18\times10^6$ | Random |
| SeCo [71] | N | 2 | 200K | 1M | 2.65×2.65 | $1.40\times10^6$ | Random |
| BigEarthNet [72] | N | 1 | 590K | 1.2M | 1.20×1.20 | $8.50\times10^5$ | Targeted |
| SatlasPretrain [73] | N | 1** | 856K | N/A | 5.12×5.12 | $2.13\times10^7$ | None |
| **M3LEO** | Y | 3 | 1.05M | 17.2M | 4.48×4.48 | $2.11\times10^7$ | None |

[*] Heuristic only — some work, such as SSL4EO-S12, considers acquisitions at the same location from different seasons to be a seperate data chip.

[**] Contains additional historical images from 2016-2021 that are relevant to dynamic events such as floods.

## 3 Dataset & Framework

### 3.1 SAR Datasets

The many benefits of SAR data are met with increased complexity compared to optical data. SAR sensors are active — that is, they emit microwave pulses (5.6 cm wavelength for Sentinel-1) and measure backscatter rather than imaging the Earth under passive illumination from the Sun. This enables day-night operation and the penetration of atmospheric obstructions such as cloud and dust. Unlike sunlight, the pulses emitted by SAR sensors are polarised, with the ability to emit pulses polarised either horizontally or vertically with respect to the Earth. SAR sensors are also able to measure the polarization of the backscatter, and capture both amplitude and phase. This signal, typically stored as a complex number, allows studying the geometry of surface-level objects, in addition to their reflectances. As an example, higher amplitudes are measured when surface features align with the polarisation of the emitted pulse. As with optical imagery, objects smaller than the measurement wavelength are invisible to the sensor.

The use of SAR data is further complicated by the use of side-looking radar. SAR sensors operate with the emitter and receiver aimed laterally, rather than vertically, as for optical sensors. Since radar operates by measuring the time of arrival for a backscattered signal, aiming the sensor vertically would make it impossible to distinguish between targets at an equal distance to the left or right of the direction of travel. This is corrected by side-looking, with the sensor aimed laterally such that the entire field of view is to one side of the satellite. Although side-looking corrects directional ambiguity, it necessitates complex post-processing to correct the resulting geometrical distortions and

radiance redistribution [81–83], and results in the same terrain being imaged differently depending on the direction of satellite travel, as the terrain is illuminated from the opposite side. We provide data in both ascending (northwards) and descending (southwards) satellite directions in M3LEO.

We provide three products derived from SAR data in this dataset — polarimetric amplitude, intereferograms, and coherence. We give a brief background on each of these datatypes below, although we omit most technical details regarding their construction as they are beyond the scope of this work. References are provided for users who are interested in further background on these datatypes.

**Amplitude**  Polarimetric amplitude measures the power of the backscattered signal received by the sensor. We provide amplitude data derived from ESA Sentinel-1 Level 1 Ground Range Detected SAR data (`S1GRD`), as available in Google Earth Engine (GEE)[1]. Phase information is not provided via GEE and it is therefore impossible to produce further SAR datatypes from data available on the GEE platform. We provide data measuring vertically polarised and horizontally polarised returns from a vertically polarised emission (referred to as VV and VH respectively). Data is provided for imagery from both ascending and descending trajectories. We refer users to [84] for a more detailed breakdown of the theory behind polarimetric SAR data. `S1GRD` data is of 10 m/pixel resolution and provided for 2018-2020 as four seasonal averages per-year.

**Interferometry**  Interformetry measures the phase difference between pairs of acquisitions over the same terrain. These phase differences provide data about small-scale displacements, which can be measured modulo the wavelength. Post-processed interferograms are *unwrapped* by computing accumulated modulo-wavelength displacements. As the scale at which these displacements can be measured depends on wavelength, rather than horizontal resolution, interferometric phase difference can be used to measure surface height changes with millimetre accuracy. We provide interferometric data computed using select pairs of Sentinel-1 acquisitions, from ASF ARIA Geocoded UNWrapped Inteferogram data (`GUNW`), as available in ASF vertex[2][85]. We include all available acquisition pairs with time deltas of less than 72 days. We refer users to [86] for a detailed treatment of the processing steps required to construct interferograms in addition to the underlying physics. `GUNW` data is provided for 2020 at a resolution of approximately 90 m/pixel.

**Coherence**  The coherence of SAR imagery is calculated using the complex correlation between coincident pixels across two separate acquisitions. For a given complex valued pixel $z^{(i)}$ in SAR acquisitions at times 1 and 2, the coherence $\gamma$ is defined as:

$$\gamma = \frac{\left| \mathbb{E}[\bar{z}_1^{(i)}(t) z_2^{(i)}(t)] \right|}{\sqrt{\mathbb{E}[|z_1^{(i)}(t)|^2] \mathbb{E}[|z_2^{(i)}(t)|^2]}} \tag{1}$$

To provide meaningful coherence values, expectations are computed within a small spatial window surrounding each pixel. The resulting resolution of coherence maps is therefore lower than that of the original acquisitions. Man-made structures typically exhibit high coherence, as they are stable across acquisitions. Forests and other vegetation larger than the wavelength of the instrument have lower coherence. Coherence is affected in all cases by additional decorrelation factors not relating directly to terrain, such as doppler centroid difference or thermal noise. [87] and [88] provide more detailed treatments of coherence estimation. We provide coherence estimates from the Global Seasonal Sentinel-1 Interferometric Coherence (`GSSIC`) dataset [89], using date-pairs with time deltas of 12, 24, 36 and 48 days, with one date-pair per tile per season. `GSSIC` data is of approximately 90 m/pixel resolution and from 2020. A decay model is included with the `GSSIC` coherence data.

For both interferometry and coherence, the selection of acquisition pairs is critical. The acquisition pair selection process introduces a significant combinatorial challenge to managing SAR datasets — if every possible combination between all acquisitions were considered, the number of interferograms or coherence estimates would grow quadratically with the number of acquisitions. We pre-empt this issue by the provision of pre-selected date-pairs.

---

[1]`developers.google.com/earth-engine/datasets/catalog/COPERNICUS_S1_GRD`
[2]`asf.alaska.edu/datasets/daac/aria-sentinel-1-geocoded-unwrapped-interferograms/`

**Optical data**   We additionally provide data (`S2SRM`) sourced from the ESA Sentinel-2 mission[3][48], as available in Google Earth Engine. Data is provided from the L2A product (surface reflectance). We do not include top-of-atmosphere L1C reflectances. We summarize this data as monthly means of cloud-free pixels. We include four monthly averages for 2018-2020 — March, June, September and December. We include all Sentinel-2 bands with a of 10 m/pixel (red, green, blue, NIR) or 20 m/pixel (vegetation red edge, SWIR 11/12).

## 3.2   Auxiliary Datasets

**ESA World Cover**   Land cover classification labels (semantic segmentation) were obtained from the ESA World Cover product (`ESAWC`) [90], as available in Google Earth Engine[4]. The resolution of `ESAWC` is 10 m/pixel, and it comprises 11 classes (See Appendix E). The `ESAWC` product has been independently validated as having an accuracy of approximately 75% [91]. Data is provided for 2020.

**ESA CCI Biomass**   A map of above ground biomass (`AGB`) in Mg ha$^{-1}$, derived from the ESA Climate Change Initiative's Biomass product[5] [92] is provided for 2020 at a resolution of 90 m/pixel. The relative error of this data is 20% for areas with a measured biomass exceeding 50 Mg ha$^{-1}$ [93]

**MODIS Vegetation Cover**   Tree cover, non-tree cover and non-vegetated (bare) percentage labels derived from the Terra MODIS Vegetation Continuous Fields product (`MODISVEG`)[6][94], are provided at a resolution of 250 m/pixel. A limited amount of independent validation reports the RMSE of the `MODISVEG` data as approximately 10% [94]. Our dataset includes yearly maps for 2016-2020.

**GHS Built Surface**   Maps of built surface area (m$^2$/pixel), derived from the Copernicus Global Human Settlement Built Surface (`GHSBUILTS`) product[7][95] are provided at 100 m/pixel for 2020. The mean absolute error of this data has been evaluated using independent reference data has been estimated to be approximately 6%[96] (or 600 m$^2$/pixel, at 100 m/pixel).

**SRTM Digital Elevation Maps**   Digital Elevation Maps, derived from the NASA Shuttle Radar Topopgraphic Mission (`SRTM`)[8][97, 98] are provided at a resolution of 90 m/pixel. This data was measured in 2000, but we emphasise that terrain height changes relatively little at this resolution compared to other products such as `ESAWC` or `GHSBUILTS`. The RMSE of the `SRTM` data was originally reported as 16 m [99] although it has been measured as more accurate in some regions [100].

## 3.3   Data coverage

Our dataset covers six distinct geographic areas of interest (AOIs): the contiguous United States (CONUS), Europe, the Middle East, Pakistan and India (PAKIN), China, and South America. A visualisation is provided in Figure 1. We limit coverage to these regions for reasons relating to the acquisition parameters used by Sentinel-1.

Firstly, Sentinel-1 operates with different acquisition modes depending on the region. We choose to only include areas where Sentinel-1 uses the Interferometric Wide (IW) swath acquisition mode. In polar regions, Sentinel-1 employs the Extra Wide (EW) swatch acquisition mode, which introduces systematic differences in how terrain is illuminated — such as the azimuth steering angle of the radar emitter being 0.8° for EW and 0.6° for IW acquisition. The polarisations used in polar regions are reversed (HH, HV vs. VV, VH) compared to other areas. We provide IW data with both VV and VH polarisations in the initial release of this dataset.

Secondly, much of the terrestrial surface of the Earth is only covered by a single direction of satellite travel. This includes much of North America, Africa, continental Asia, Oceania and the Amazon rainforest. Unlike passively illuminated data such as Sentinel-2, SAR actively illuminates terrain with

---

[3]`developers.google.com/earth-engine/datasets/catalog/sentinel-2`
[4]`developers.google.com/earth-engine/datasets/catalog/ESA\_WorldCover\_v200`
[5]`gee-community-catalog.org/projects/cci_agb/`
[6]`developers.google.com/earth-engine/datasets/catalog/MODIS\_006\_MOD44B`
[7]`human-settlement.emergency.copernicus.eu/ghs_buS2023.php`
[8]`developers.google.com/earth-engine/datasets/catalog/CGIAR_SRTM90_V4`

a radar emitter aimed laterally from one side of the satellite. Orbital direction therefore systematically determines whether terrain is illuminated from an easterly or westerly direction.

By focusing on regions with consistent acquisition parameters, we aim to provide a dataset that is more uniform and suitable for training models without introducing additional complexities. While including other regions might reduce geographic bias, it is not straightforward to address the potential systematic biases introduced by varying acquisition modes and polarisations, or the systematic lack of directional coverage in some regions. These issues warrant a detailed treatment beyond the scope of this work, and we refer readers to the Copernicus Wiki for a full set of details on Sentinel-1 coverage[9] (particularly Figures 20, 21). Processing additional coverage is technically straightforward using the provided tools, should it be required. We include data for Europe despite its absence of GUNW coverage due to interest from data providers operating in Europe.

A total of 1,048,827 unique geographic tiles were generated, covering an area of $2.11 \times 10^7$ km$^2$. A breakdown by AOI can be seen in Table 2. Tiling is uniform across all AOIs and datasets—the chips provided for each dataset cover exactly the same geographical areas. It is important to note that not all component datasets or specific parameterisations, such as date-pair ranges, were available for every geographic tile; however, the availability is still consistently high. See Appendix A for a full breakdown by dataset and AOI. We provide a reduced version of our dataset, **M3LEO-miniset**, spanning 5,000 tiles per AOI for rapid model iteration and use in tutorials.

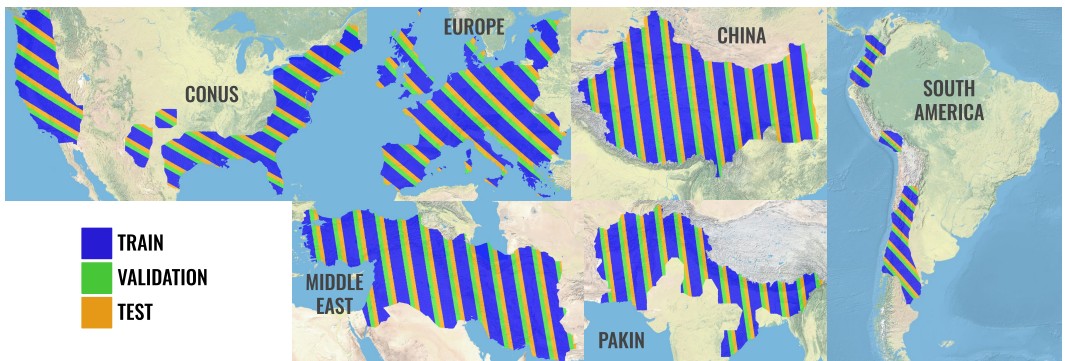

Figure 1: **Data splits**. Geographic bands for training, validation and test sets, at a ratio of 60:20:20.

Table 2: **Coverage statistics** summarised for each AOI in M3LEO.

| AOI | Total No. Tiles | Area (km$^2$) | % Earth's land surface |
|---|---|---|---|
| CONUS | 167403 | $3.360 \times 10^6$ | 2.3% |
| Europe | 200489 | $4.024 \times 10^6$ | 2.7% |
| Middle East | 163986 | $3.291 \times 10^6$ | 2.2% |
| PAKIN | 147791 | $2.966 \times 10^6$ | 2.0% |
| China | 285402 | $5.728 \times 10^6$ | 3.7% |
| South America | 83756 | $1.681 \times 10^6$ | 1.1% |
| **Total** | **1048827** | **21.05$\times 10^6$** | **14.1%** |

**Data splits** We provide use geographic bands to define training, validation and test splits for use in the explorations foudn in this work, following [65]. Bands were split into training, validation and test sets at a ratio of 60:20:20, and can be seen visually in Figure 1. This method reduces distribution shifts and data leakage compared to single-band splits and fully randomized assignments, respectively. See Appendix A for details. Users should define train-test splits appropriate for their individual applications if their needs differ.

---

[9]sentiwiki.copernicus.eu/web/s1-mission

### 3.4 Framework

**Data downloading & processing**  For each AOI, we provide a `.geojson` file containing the geographical extent and unique ID for each tile. These definitions are applied to each dataset at the point of tiling such that any chip with a given identifier spans precisely the same geographic extent of a chip corresponding to a different dataset of the same identifier. Throughout this work, we use the term *tile* to refer to a fixed area of the Earth's surface defined in the definition files, and the term *chip* to refer to the data from a single component dataset, such as `S1GRD` or `ESAWC`, within the extent of a given tile.

To process datasets available via Google Earth Engine (GEE) (`S2RGB`, `S1GRD`, `SRTM`, `ESAWC`, `MODISVEG`), we introduce `geetiles`[10]. This tool extracts and tiles data from GEE as per definition files and configurations provided in the `geetiles` repository. The remaining datasets (`GSSIC`, `GUNW`, `AGB`, `GHSBUILTS`) were extracted using `sartiles`[11], which contains specific code to download and tile each of `GUNW` and `GSSIC`, as well as the facility to tile general GeoTIFF files, such as those provided for `AGB` and `GHSBUILTS`, for integration with M3LEO. Both `geetiles` and `sartiles` were developed alongside M3LEO, and are provided such that users are able to seamlessly integrate any data available via Google Earth Engine (or as a GeoTIFF file) with our framework.

**Pipeline**  We accompany our dataset with a modular PyTorch Lightning [101] framework parameterised using Hydra [29]. We provide PyTorch Lightning datasets for each component of M3LEO. We also provide additional modules defining self-supervised approaches applied successfully to M3LEO in previous works (MAE [65], CLIP [67], DINO [68, 69]). Integrating custom models with M3LEO is straightforward, requiring the addition of a single file.

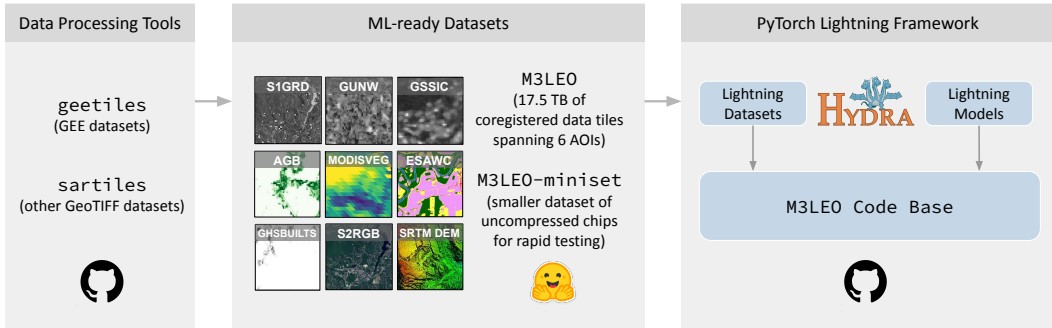

Figure 2: **M3LEO dataset and framework.** The M3LEO dataset consists of nine ML-ready component datasets and a PyTorch Lightning framework, parameterised by Hydra, for model training.

## 4 Analysis

We provide five auxiliary datasets (`ESAWC`, `AGB`, `MODISVEG`, `GHSBUILTS`, `SRTM`) in addition to SAR and optical satellite data acquisitions. Although these data could be used as labelled tasks (see Appendix D, and also [65, 67–69] for analysis of this type on M3LEO), they could equally be considered as providing information on important surface properties that would be non-trivial to derive directly from satellite data.

We compare the shift in the marginal distribution of these terrain properties $y$, $p(y)$ in Figure 3 for four of these auxiliary datasets. We computed the normalised $L_1$ distance between the discrete `ESAWC` distributions and the Wasserstein distance between the continuous `GHSBUILTS`, `MODISVEG` and `SRTM` distributions, across AOIs. The marginal distributions $p(y)$ for each auxiliary dataset are shown in Appendix B.

We also provide an early exploration of the 'appearance shift' of features between AOIs for `S1GRD` — the change in the distribution of embeddings $x$ for a SAR tile with known terrain properties $y$, $P(x|y)$.

---

[10] github.com/rramosp/geetiles
[11] github.com/rramosp/sartiles

To generate embeddings, we trained a masked autoencoder on `S1GRD` polarimetry from all six AOIs, following previous work on M3LEO [65]. Training hyperparameters can be seen in Appendix C. We applied max-pooling along the sequence dimension at the output of the ViT encoder and further reduced the dimension to 2 using UMAP [102]. We computed the expected Wasserstein distance between the conditional distributions $p(x|y)$ with respect to the distribution $p(y)$ on the test set, across AOIs and show the results in Figure 4. For ESAWC, we applied principal component analysis and conditioned on the first 3 principal components with 10 evenly spaced bins per dimension. On the remaining variables, we used 100 evenly spaced bins. We also display the sliced Wasserstein distance between the embedding distributions $p(x)$ in the leftmost matrix of Figure 4. We plot these reduced embeddings, coloured according to each of the terrain properties, in Figure 5. A checkpoint for the MAE model is available in the data repository.

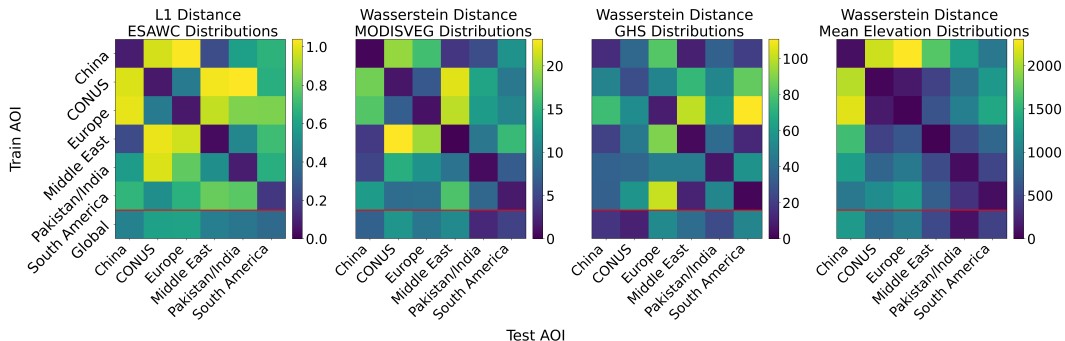

Figure 3: **Distribution shifts**. Distribution shifts between terrain properties described by auxiliary datasets. Distribution shift for the ESAWC categorical data was quantified using L1 distance and continuous data using Wasserstein distance.

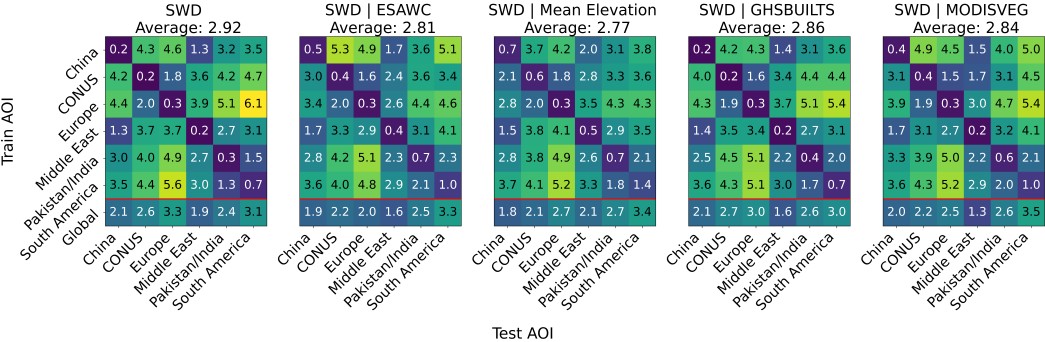

Figure 4: **Covariance and appearance shift**. **(Leftmost)** The Sliced Wasserstein Distances (SWD) between reduced train and test set embeddings across AOIs, generated using a masked autoencoder. **(Right four)** The expected value of same metric conditioned on each of four terrain properties. The expectation was computed with respect to the distribution of the terrain property on the test set. We conditioned on the first three principal components of the `ESAWC` distribution.

We observed that there was significant covariate shift in the distribution of the embeddings produced by the masked autoencoder, $p(x)$, across AOIs (Figure 4, leftmost matrix). Given that there was also significant shift in terrain properties described by the auxiliary datasets between AOIs (Figure 3, this is not immediately surprising. Another factor that must be considered, however, is the shift in the embeddings produced by the masked autoencoder for tiles with similar terrain properties $y$, $p(x|y)$, across AOIs (appearance shift). It can be seen in the four right-hand matrices of Figure 4 that although there is some reduction in the most extreme cases, the expected value of sliced Wasserstein distance conditioned on similar terrain properties is usually not substantially lower than the unconditioned case. Explained from an Earth observation perspective - the masked autoencoder does not extract overly similar features for two tiles with, for example, very high vegetation when those tiles are taken from different geographic regions. This effect can be seen visually in Figure 5 - embeddings with

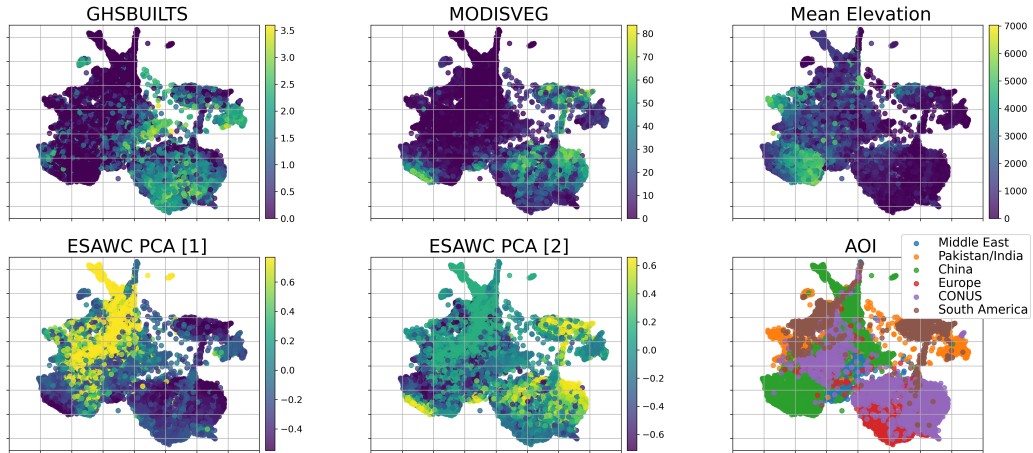

Figure 5: **Embedding scatter plots**. Scatter plots of 2D embeddings, reduced using UMAP, coloured according to different auxiliary datasets.

high values for particular labels do not cluster obviously across different AOIs. In contrast, previous work applying DINO-based self-supervision to M3LEO [69] found that embeddings with similar labels clustered tightly in the embedding space across AOIs. Despite this observation, previous work applying MAE-based pretraining to M3LEO has shown strong generalisation to novel AOIs [65].

We did not explore the use of `GSSIC` coherence data here, as this is substantially technically challenging and requires detailed treatment. Given the lower resolution of this data, it may not be productive to use as a naive input to deep learning models. A small set of experiments evidencing this is provided in Appendix D. We point readers to previous work using coherence data from M3LEO productively in a self supervised setting - in contrastive learning [67] or in knowledge distillation-based approaches [69]. We note that the provision of `GUNW` interferometric coherence data without reference to specific events such as floods or fires is unusual compared to other datasets [76–78]. Although we aim to include event-based datasets in a future update, this large-scale dataset of interferograms is still highly desirable in self-supervised schemes.

A number of unknowns remain regarding domain shift in M3LEO. Many auxiliary datasets, such as `ESAWC`, do not exist for 2018 and 2019 so it is difficult to provide a substantial exploration of temporal shifts, although we provide `S1GRD` and `S2SRM` for three years. It is unclear whether features learned from encoders trained on different polarizations or orbital directions are comparable, although D contains a limited set of experiments on the value-add of different polarisations.

## 5    Conclusions and Future Work

In this work, we introduced M3LEO, a multi-continental Earth observation dataset including a comprehensive set of SAR data, alongside digital elevation models, RGB data and a suite of downstream tasks. To the best our knowledge, this is the largest ML-readable polSAR dataset by total number of tiles and geographic coverage, and the largest inSAR dataset by the same metrics. We additionally provide a modular PyTorch Lightning framework to enable the application of deep learning and encourage the uptake of these datatypes. We provide additional tools, `geetiles` and `sartiles`, to enable the integration of any data available in Google Earth Engine with our framework.

We trained an MAE-based model on polSAR data and conducted a small exploration on the appearance shift of features corresponding to similar labels across AOIs. Despite the fact that this type of training has previously been shown to generalise well geographically [65], the shift in low-level features useful for the reconstruction pretext task was substantial between geographic regions. This is in contrast to previous work using M3LEO that found embeddings from DINO-based models with similar labels from different AOIs clustered tightly.

# 6 Acknowledgements

This work has been enabled by Frontier Development Lab Europe (`https://fdleurope.org`) a public / private partnership between the European Space Agency (ESA), Trillium Technologies, the University of Oxford and leaders in commercial AI supported by Google Cloud and Nvidia, developing open science for all Humankind. L.M-F. was supported by the European Research Council (ERC) Synergy Grant "Understanding and Modelling the Earth System with Machine Learning (USMILE)" under the Horizon 2020 research and innovation programme (Grant agreement No. 855187). M. J. A. was supported by the UKRI Centre for Doctoral Training in Application of Artificial Intelligence to the study of Environmental Risks [EP/S022961/1]. We are also indebted to Nicolas Longépé, Carlos López-Martínez, Fabio A. González Osorio, Samuel Bancroft, Emma Hatton, Alison Lowndes, Alistair Francis, Ioanna Bouri and the rest of reviewers during the 2023 FDL-Europe sprint.

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

# A  Dataset Summary

## A.1  Chip counts

Table A.1: **Summary of component datasets** in M3LEO, including number of chips and dataset size for each region (per year). Totals at the bottom are adjusted for multi-year datasets.

| Input Datasets | CONUS | | Europe | | China | |
|---|---|---|---|---|---|---|
| | **Chips** | **Size/GB** | **Chips** | **Size/GB** | **Chips** | **Size/GB** |
| S1GRD (2018-2020) | 167403 | 1003 | 200489 | 1228 | 285402 | 1740 |
| GSSIC (2020) | 167403 | 73 | 200489 | 104 | 285399 | 125 |
| GUNW (2020) | 554844 | 579 | N/A[*] | N/A[*] | 1027451 | 854 |
| S2SRM (2018-2020) | 167406 | 1228 | 200489 | 1433 | 285402 | 1945 |
| ESAWC (2020) | 167406 | 32 | 200489 | 39 | 285402 | 55 |
| AGB (2018-2020) | 167406 | 8.5 | 200489 | 11 | 285402 | 14 |
| MODISVEG (2020) | 167406 | 151 | 200489 | 189 | 285402 | 220 |
| GHSBUILTS (2020) | 167406 | 0.7 | 200489 | 1.5 | 285402 | 2.4 |
| SRTM (2000) | 167406 | 126 | 200489 | 151 | 285402 | 215 |
| **Total** | **2,563,704** | **7,680.2** | **2,806,846** | **8,500.5** | **5,023,076** | **12568.4** |

| Input Datasets | Middle East | | PAKIN | | S. America | |
|---|---|---|---|---|---|---|
| | **Chips** | **Size/GB** | **Chips** | **Size/GB** | **Chips** | **Size/GB** |
| S1GRD (2018-2020) | 163986 | 983 | 147791 | 886 | 83756 | 502 |
| GSSIC (2020) | 158985 | 68 | 147791 | 69 | 83756 | 34 |
| GUNW (2020) | 608865 | 619 | 486914 | 309 | 226093 | 155 |
| S2SRM (2018-2020) | 163986 | 1126 | 147791 | 992 | 83756 | 529 |
| ESAWC (2020) | 163986 | 32 | 147791 | 29 | 83756 | 16 |
| AGB (2018-2020) | 163986 | 7.7 | 147791 | 7 | 83756 | 4.1 |
| MODISVEG (2020) | 163986 | 88 | 147791 | 113 | 83756 | 79 |
| GHSBUILTS (2020) | 163986 | 1.3 | 147791 | 1.2 | 83756 | 0.7 |
| SRTM (2000) | 163986 | 124 | 147791 | 112 | 83756 | 63 |
| **Total** | **2,899,668** | **7,282.4** | **2,555,988** | **6,288.2** | **1,398,677** | **3,453** |

[*] GUNW data unavailable for Europe.

## A.2  Banding

As outlined in Section 3.3, we stratified our data into training, validation and test splits using geographic banding. 60 bands were constructed at a fixed orientation for each AOI. See Table A.2 for the angles used per-AOI. Specifically, we allocated three adjacent bands for the training set, followed by one adjacent band each for the validation and test sets, in sequence, until all bands were categorized. See Figure 1 for a visual representation.

Table A.2: **Angles** for the construction of training, validation and test splits by geographic banding. Angles are measured in radians, clockwise (axis pointed towards the Earth), with west corresponding to an angle of 0.

| AOI | Band Angle |
|---|---|
| CONUS | 0.9 |
| Europe | 0.9 |
| China | 1.5 |
| Middle East | 1.5 |
| PAKIN | 1.5 |
| South America | 0.6 |

# B  Marginal Distributions

The marginal distributions of `ESAWC`, `MODISVEG`, `STRM` mean elevation and `GHSBUILTS` can be seen in Figures B.1, B.2, B.3 and B.4 respectively.

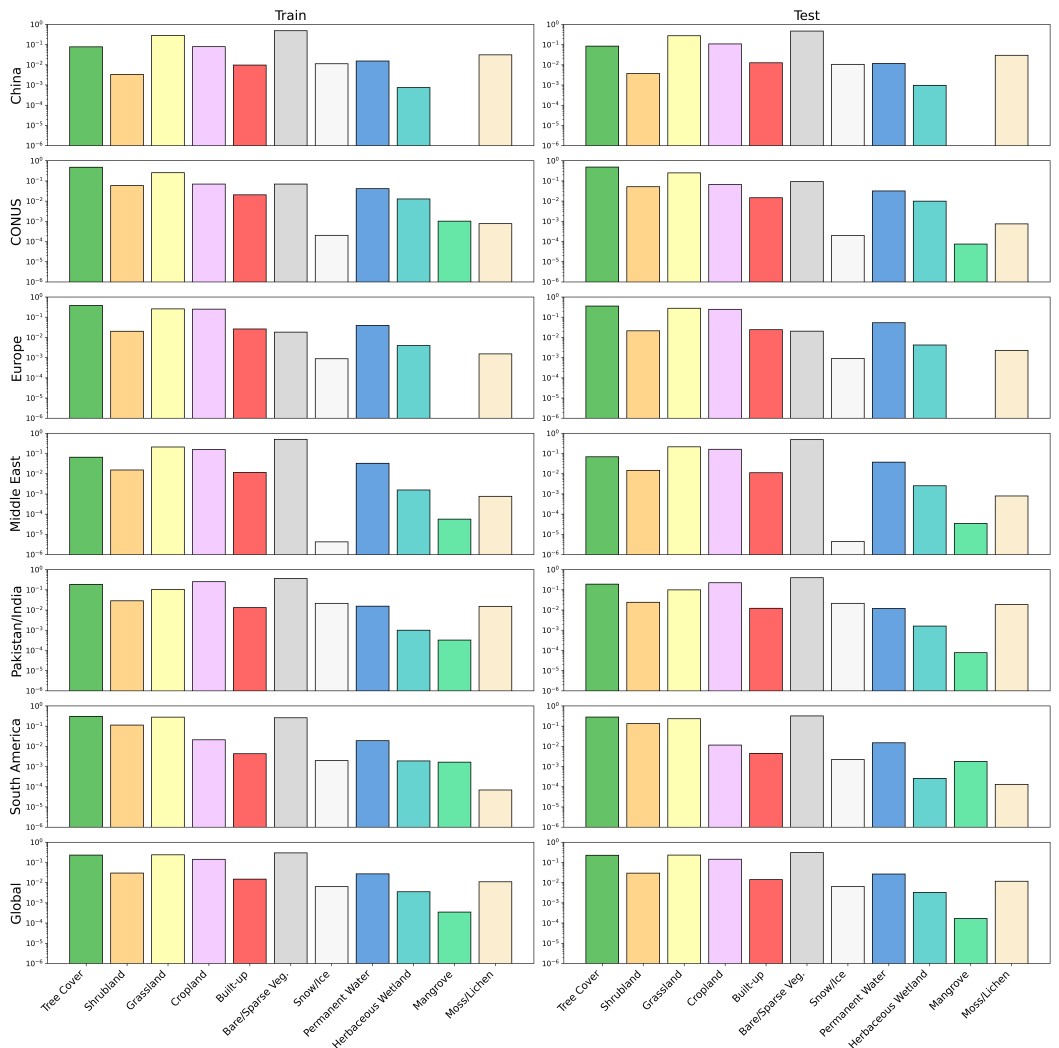

Figure B.1: **Marginal distribution for** `ESAWC`. Note log scale.

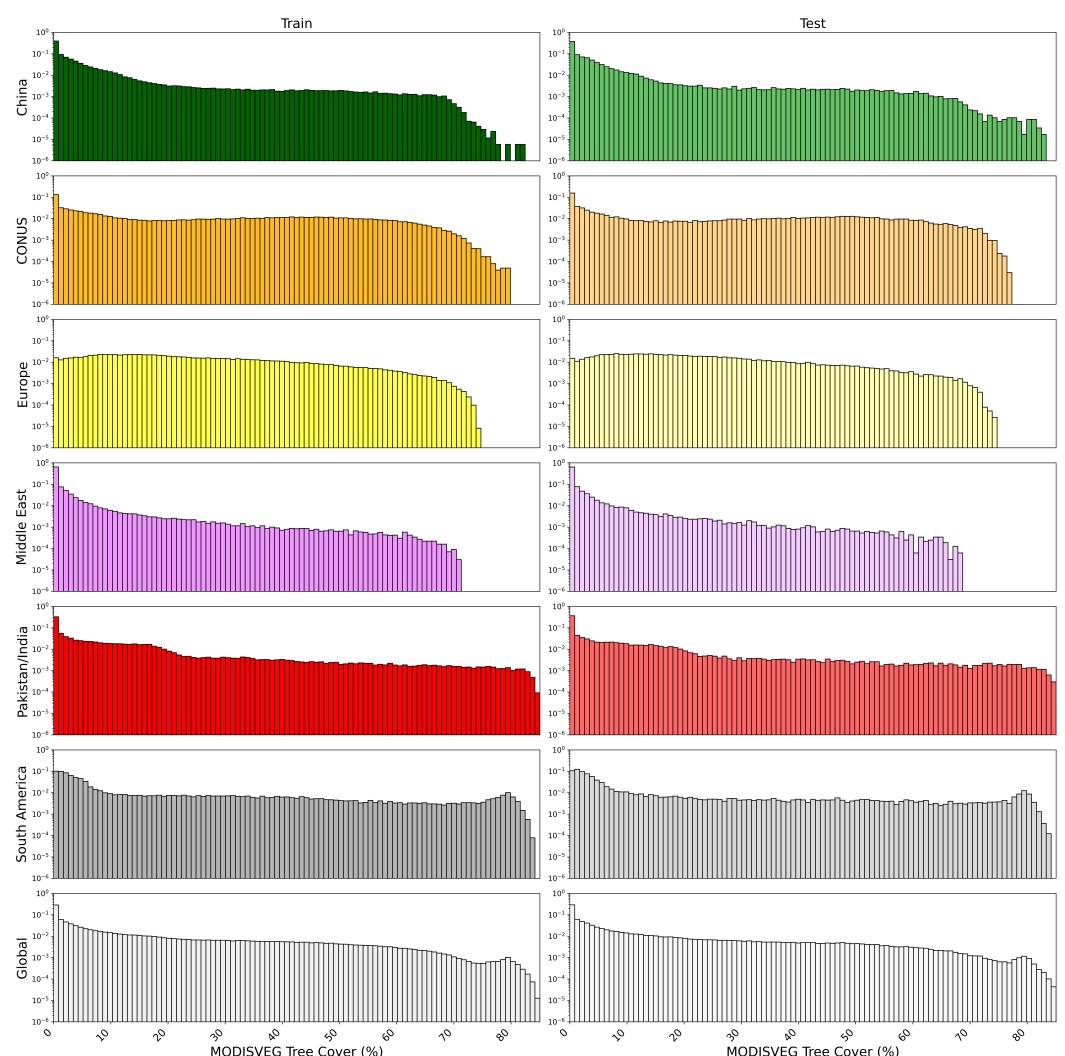

Figure B.2: **Marginal distribution for** `MODISVEG`. Note log scale.

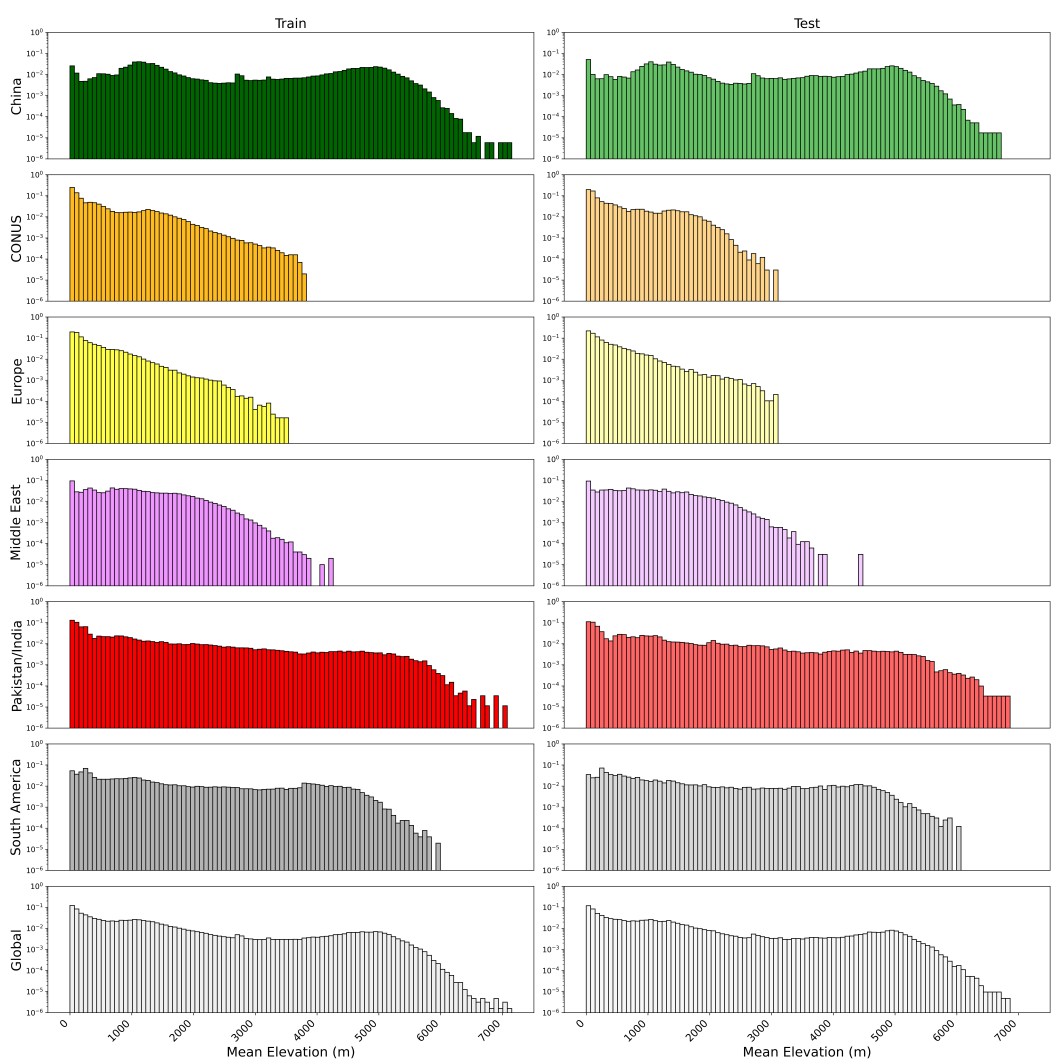

Figure B.3: **Marginal distribution for** SRTM. Mean elevation per-chip. Note log scale.

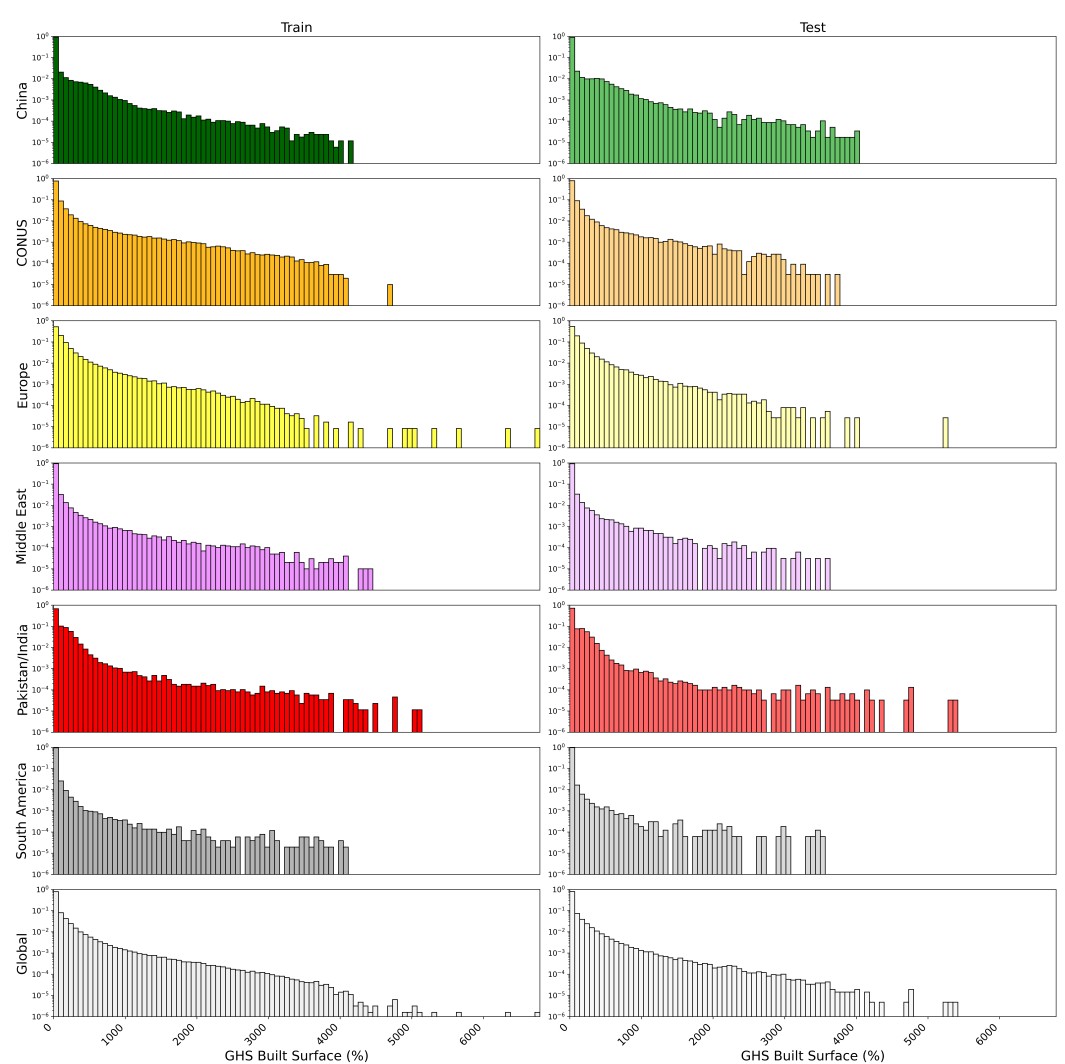

Figure B.4: **Marginal distribution for** GHSBUILTS. Note log scale.

# C MAE Hyperparameters

Training hyperparameters for the MAE-based model can be seen in Table C.1. We followed [65], other than including additional AOIs in the pretraining set. A checkpoint for this model is available at `huggingface.co/M3LEO`.

Table C.1: **Hyperparameter details** for MAE pretraining used in main text.

|  | **MAE Pretraining** |
| --- | --- |
| **Encoder** | ViT-B [33] |
| (Params) | (88.8M) |
| **Decoder** | Reconstruction [35] |
| (Params) | (5.5M) |
| **Loss Function** | MSE |
| **Input Image Size** | $448 \times 448$ |
| **Output Image Size** | $448 \times 448$ |
| **Patch Size** | $16 \times 16$ |
| **Masking Type** | Random |
| **Masking Ratio** | 0.75 |
| **Optimiser** | AdamW |
| **Learning Rate** | 1.00E-04 |
| **No. Epochs** | 75 |
| **Input Dataset** | S1GRD |
| **Channels** | Seasonal, VV, VH, VV-VH, Ascending |
| **AOIs** | CONUS, Europe, China, Middle East, PAKIN, S. America |

# D Supervised Experiments

In addition to our explorations of distribution shift, we performed a small set of supervised experiments, reframing the auxiliary datasets as labelled tasks. S1GRD, GSSIC coherence and S2SRM were used as input datasets. For S1GRD, we trained models separately using the VV and VH bands only, and with the VV and VH bands stacked at the input to the model. For all S1GRD models, four seasonal summaries were used for each band, resulting in four input channels for the VV and VH models, and eight channels for the model using both VV and VH. For GSSIC coherence, the coherence band was used with a single date pair of delta 36 days taken for each season, resulting in four input channels. For the S2SRM models, we used the red, green and blue channels with one input channel from each of the months of March, June, September and December, totalling 12 input channels. We additionally trained models using both S2SRM RGB in combination with each of the other datasets separately, stacking the bands at the input to the model. All input data was taken from 2020. We upscaled low resolution input datasets to $448 \times 448$ px before input to the model.

We excluded GUNW from use in our baseline experiments to avoid introducing the mixed availability of GUNW chips as a confounding factor.

ESAWC, AGB and GHSBUILTS labelled datasets were used as targets. ESAWC was formulated as semantic segmentation spanning 11 land use classes, for which segmentation accuracy is reported as mean intersection-over-union (mIoU). ESAWC data was used at the original resolution of $448 \times 448$. Both AGB and GHSBUILTS are formulated as regression tasks (per-pixel). Results are reported using root mean square error (RMSE), in Mg ha$^{-1}$ for AGB and in m$^2$ built surface for GHSBUILTS. We resized labels for AGB and GHSBUILTS to a fixed size of $45 \times 55$, to account for minor differences in dimension from chip-to-chip. We note that this means our pixels may not span exactly 1 hectare and therefore that results for AGB measured in Mg ha$^{-1}$ are a heuristic rather than an absolute measure of biomass.

We excluded data from Europe in these experiments due to the absence of GUNW data in this region. This constraint was applied despite having not used GUNW in these baselines, as these experiments were completed prior to the exclusion of GUNW and repeating them was prohibitively expensive. All models were trained on the same random $10\%$ subset of the data. The spatial coverage of this subset is still similar to other popular large-scale EO datasets. We used the entire test set to compute the final metrics.

**Models**  We used a UNet-style architecture for all baseline experiments, following [103] with two major changes — halving the number of channels for all layers and replacing the up-convolutions with bilinear upsampling. For the AGB and GHSBUILTS regression tasks, we applied average pooling to the single-channel $448 \times 448$ UNet output to achieve an output dimension of $45 \times 55$. We opted not to use data augmentation. Selecting augmentations for SAR data is challenging — common choices such as rotation or flipping may introduce invariance to information specific to different instrument polarisations or orbital direction, for example. For further details on training and hyperparameters, see Appendix D.3.

## D.1 Results

Results for all tasks are reported in Table D.1. For the experiments using a single data source as input, S1GRD using both polarisations (VV+VH) achieved the best result for the ESAWC (MIoU: 0.4185) and AGB (RMSE: 27.467 Mg ha$^{-1}$) tasks. S2RGB achieved the best result for GHSBUILTS (RMSE: 131.968 m$^2$). GSSIC coherence produced the worst result in all three cases (ESAWC MIoU: 0.2906, AGB RMSE: 39.314 Mg ha$^{-1}$, GHSBUILTS RMSE: 131.968 m$^2$). The best results for experiments using multiple data sources as input were achieved by fusion of S1GRD and S2RGB in all cases (ESAWC MIoU: 0.4634, AGB RMSE: 25.137 Mg ha$^{-1}$, GHSBUILTS RMSE: 124.852 m$^2$), with significant improvements over either S1GRD or S2RGB individually. Fusion of S2RGB with GSSIC improved results marginally (ESAWC MIoU: 0.4198, AGB RMSE: 28.550 Mg ha$^{-1}$, GHSBUILTS RMSE: 131.300 m$^2$) compared to either individual data source individually in all cases.

## D.2 Discussion

In all experiments, S1GRD combining VV and VH polarisations performed similarly to S2RGB, although the gap was small. The inclusion of both polarisations for S1GRD increased performance

Table D.1: **Baseline Evaluation Results** for `ESAWC`, `AGB`, and `GHSBUILTS` tasks using our UNet models, outlined in Section D.

| Input Dataset | Bands | ESAWC MIoU | AGB RMSE (Mg ha$^{-1}$) | GHSBUILTS RMSE (m$^2$ built) |
|---|---|---|---|---|
| S1GRD | VV | 0.3976 | 28.573 | 152.259 |
| S1GRD | VH | 0.3787 | 30.333 | 152.847 |
| S1GRD | VV+VH | **0.4185** | **27.467** | 141.719 |
| GSSIC | Coherence | 0.2906 | 39.314 | 196.242 |
| S2RGB | RGB | 0.4094 | 28.689 | **131.968** |
| S2RGB+S1GRD | RGB+VV+VH | **0.4634** | **25.137** | **124.852** |
| S2RGB+GSSIC | RGB+Coherence | 0.4198 | 28.550 | 131.300 |

uniformly compared to either individually. Unlike optical data, reflected SAR pulses are polarised differently depending on the terrain, and each measured polarisation contains unique information about the geometry of the imaged area. In all cases, performance was significantly improved by using both `S1GRD` and `S2RGB` data in fusion, despite giving similar performances individually, confirming a common finding in previous work.

Baseline experiments using `GSSIC` as the sole input data source performed significantly worse than for other input data types. This is likely explained in large part by the difference in resolution between `GSSIC` (90m, upsampled to 10m), `S1GRD` (10m) and `S2RGB` (10m). Performance improved slightly in all cases when `GSSIC` was included alongside in fusion with `S2RGB`. A nuanced approach is required for including coherence and interferometry, rather than naive direct input. Users may wish to use these data sources in self-supervised learning schemes — for example, pretraining models by constructing coherence or interferometry data from polarimetry pairs, or in a contrastive scheme alongside polarimetry data. Some work has been successful in using this type of pretraining [67]. Motivated by the success of polarimetry and coherence data in non-ML tasks [13–20], we suggest that these data types generated from date-paired SAR acquisitions are likely to show stronger performance on change detection-type tasks, which we did not demonstrate here. M3LEO may provide useful pretraining data for these tasks.

### D.3 Training Hyperparameters

Hyperparameter details for supervised experiments can be seen in Table D.2. We did not perform significant hyperparameter tuning. Model selection was using best performance on the validation set. All models used approximately $7.9 \times 10^6$ trainable parameters, to the nearest $10^5$. We show runtimes for each experiment on 2 NVIDIA V100 GPUs in Table D.3.

Table D.2: **Hyperparameter details** for supervised experiments

| | ESAWC | AGB | GHSBUILTS |
|---|---|---|---|
| **Task Type** | Semantic Segmentation | Regression | Regression |
| **Loss Function** | Cross Entropy | RMSE | RMSE |
| **Input Dimensions** | $448 \times 448$ | $448 \times 448$ | $448 \times 448$ |
| **Output Dimensions** | $448 \times 448$ | $45 \times 55$ | $45 \times 55$ |
| **Output Channels** | 11 | 1 | 1 |
| **Optimiser** | Adam [104] | Adam [104] | Adam [104] |
| **Learning Rate** | 1.00E-04 | 1.00E-04 | 1.00E-04 |
| **No. Epochs** | 75 | 75 | 75 |
| **Model Selection** | Best (val) | Best (val) | Best (val) |

Table D.3: **Runtimes** for single-input and data fusion baseline experiments

| Input Dataset | ESAWC | AGB | GHSBUILTS |
|---|---|---|---|
| S1GRD (VV) | 16h 23m | 15h 38m | 15h 45m |
| S1GRD (VH) | 16h 39m | 15h 41m | 15h 48m |
| S1GRD (VV+VH) | 16h 54m | 17h 29m | 23h 50m |
| GSSIC | 16h 29m | 15h 33m | 15h 41m |
| S2 | 17h 51m | 17h 59m | 18h 2m |
| S2+S1GRD (VV+VH) | 22h 54m | 24h 35m | 23h 19m |
| S2+GSSIC | 19h 23m | 68h 20m | 19h 49m |

# E   ESA World Cover Land Cover Classes

We outline the 11 classes that together comprise the ESAWC dataset in Table E.1, along with their pixel values in M3LEO. We opted to leave these pixel values as-found in the original ESAWC.

Table E.1: **ESAWC Classes** along with pixel values for data as-found in M3LEO.

| Pixel Value | Class |
|---|---|
| 10 | Tree Cover |
| 20 | Shrubland |
| 30 | Grassland |
| 40 | Cropland |
| 50 | Built-up |
| 60 | Bare/Sparse Vegetation |
| 70 | Snow and Ice |
| 80 | Permanent Water Bodies |
| 90 | Herbaceous Wetland |
| 95 | Mangroves |
| 100 | Moss and Lichen |

# F Limitations

While we highlight that M3LEO comprises ML-ready data and and easy-to-use framework, we also call attention to a number of potential limitations regarding both the data and framework.

## F.1 Data Limitations

**Coverage**   While the M3LEO dataset is large, coverage is not global. We limited coverage to the area covered in the `GUNW` dataset, which is approximately equal to the regions in which Senintel-1 has dual-polarization, ascending-descending coverage. Generating further inteferometric data is substantially technically complex, computationally demanding and requires the use of SAR acquisitions with phase information (which are difficult to access compared to the amplitude data we provide). Were this data to be generated, cloud storage requirements for M3LEO would approach the petabyte scale, for which we are unable to provide a feasible long-term storage solution.

**Change Detection & Time Series Data**   We highlight the potential application of interferometric data to change detection tasks, but note that this data is not included in the initial relase of M3LEO. Two datasets that could be used for change detection tasks have been processed — namely, ESA CCI Burned Area MODIS[12] and the Global Flood Database[13] — but have not been tested extensively enough for inclusion here. We are unable to guarantee the release of these components simultaneously with the data advertised in the main text of this work, but aim to release them in the future.

**Multitemporal Data**   Data from `S1GRD`, `S2SRM` and `AGB` have been tiled for 2018-2020 additionally, but other datasets are provided for 2020 only. Many datasets, such as `ESAWC`, simply do not exist for 2018 or 2019. One satellite of the Sentinel-1 pair, Sentinel-1B, suffered a power unit failure in December 2021, so we cannot provide data with the same coverage as 2018-2020 from 2021 onwards.

## F.2 Framework Limitations

**Data Loading**   Data is currently loaded from the disk using `xarray`. We chose to use `xarray` as it easily accommodates handling the wealth of metadata associated with remote sensing imagery, but it is not performant for loading a large number of tiles quickly. For users who wish to access the same data many times — either over many epochs, or many experiments — we recommend caching the returned data arrays at the first encounter. We provide this facility using `blosc2`. The caching process can be computationally expensive for large datasets, but is relatively far cheaper than performing repeat runs using `xarray`. Slightly increased disk space requirements are incurred. We provide data for download as parquet files, but Apache Spark is not currently integrated with the framework and this data will need to be decompressed before use.

**Visualisation**   While we did not advertise data visualisation within the main body of this work, a small number of tools to visualise model outputs exist. We aim to include these in the initial code release but are unable to guarantee this.

**Tile Size**   We provide data chips at a fixed spatial size of $448 \times 448$ m. While we provide the facility to re-tile data straightforwardly at different sizes with `geetiles` and `sartiles`, this process is computationally expensive when tiling over large spatial areas.

**Benchmarking**   While we provided some baseline results (Appendix D) using our framework and data, we did not provide a benchmarking framework under which models could be compared. For example, we made no assertion as to whether models should be evaluated on chips with missing channels — we chose to fill any missing data with a dummy value of $0$. We also did not include European data in our test set. Although comparisons could be made by copying our approach exactly, we encourage domain experts to evaluate models according to the needs of their particular application.

---

[12]developers.google.com/earth-engine/datasets/catalog/ESA_CCI_FireCCI_5_1

[13]developers.google.com/earth-engine/datasets/catalog/GLOBAL_FLOOD_DB_MODIS_EVENTS_V1

# G   Licenses

The licenses under which each component of M3LEO was originally distributed are listed in Table G.1. **We distribute our framework and data under the Creative Commons BY-SA 4.0 license.**.

Table G.1: **Licenses** for components of M3LEO

| Dataset | License |
|---------|---------|
| S1GRD | Creative Commons BY-SA 3.0 IGO |
| GSSIC | Creative Commons BY 4.0 DEED |
| GUNW | Other (free use)[1] |
| S2RGB | Creative Commons BY-SA 3.0 IGO |
| ESAWC | Creative Commons BY 4.0 DEED |
| AGB | Other (free use)[2] |
| MODISVEG | No restrictions[3] |
| GHSBUILTS | Other (free use)[4] |
| SRTM | Other (free-use)[15] |

[1] `https://www.jpl.nasa.gov/jpl-image-use-policy`
[2] `https://artefacts.ceda.ac.uk/licences/specific_licences/esacci_biomass_terms_and_ conditions_v2.pdf`
[3] `https://developers.google.com/earth-engine/datasets/catalog/MODIS_006_MOD44B# terms-of-use`
[4] `https://eur-lex.europa.eu/legal-content/EN/TXT/?uri=CELEX:32011D0833`
[5] `https://developers.google.com/earth-engine/datasets/catalog/USGS_SRTMGL1_003# terms-of-use`

