# OpenReview forum: "M3LEO: A Multi-Modal, Multi-Label Earth Observation Dataset Integrating Interferometric SAR and Multispectral Data"
_NeurIPS.cc/2024/Datasets_and_Benchmarks_Track — NeurIPS 2024 Track Datasets and Benchmarks Poster_

### Official Review · Reviewer_eCWi · 2024-07-22
**Relevant and well-packaged dataset with potential for expanding of provided baselines**

**Rating:** 8
**Confidence:** 5
**Clarity:** The paper is very well written and ea…

**Review:**

Overall this is a strong submission that tackles an important gap in the literature: a large-scale aligned optical and radar dataset, ready for ML-use. My biggest concern is one that I believe can be addressed and that would, in my opinion, elevate the paper further. This dataset could be very interesting to evaluate pretrained EO foundation models and assess spectral scaling laws - performance changes across different bands used. Including functionalities to either fine-tune or use embeddings of latest generation EO foundation models such as Prithvi or Clay into the repository would make this a very valuable resource for the ML + EO communities.

**Strengths:**

This paper provides an extensive and comprehensive Earth observation dataset that, in my opinion, tackles an important gap in the literature: SAR data is currently often overlooked in the ML/CV communities in favor of optical imagery which naturally fits better with many existing computer vision frameworks and pretrained models. Nonetheless, as the authors lay out, there are many benefits to radar data, including its ability to e.g. "see" through clouds. The dataset is designed and packaged well. I especially appreciate the authors efforts to include geographic splits, a practice essential to naturally autocorrelated geographic data. Lastly, I appreciate the small but interesting baseline evaluation section. Particularly the discussion section provides some interesting insights and poses that , at least for the low-resolution benchmark tasks at hand, optical and radar data perform somewhat similarly.

**Additional Feedback:**

N/A

**Correctness:**

The paper appears correct to me. Some choices, like e.g. the design of the spatial splits, seem somewhat arbitrary to me but not wrong.

**Documentation:**

The code appears to be comprehensive and hands-on example notebooks are provided to help users to get started.

**Ethics:**

No ethical concerns with these kind of spatial resolutions.

**Limitations:**

The authors address limitations well, especially concerning the nature, advantages and disadvantages of SAR data.

**Opportunities For Improvement:**

I have not many issues with the paper but want to highlight a few things:

The clear major shortcoming for me is that, while the dataset is great, it should have been tested with more existing and publicly accessible downstream models. While the authors train their own UNet model to test performance of different bands, no existing pretrained EO foundation model is tested. This is quite a low-hanging effort considering the broad availability of EO foundation models that can process both optical and radar data such as Prithvi and Clay. I would have been very curious to see how these models (either fine tuned or their embeddings used) compared to the trained-from-scratch UNet the authors provide. Especially if models are not fine-tuned but embeddings are used, this also does not necessarily constitute an extensive computational effort. I would strongly encourage the authors to include such baselines in the final version of their manuscript and believe it would improve the submission.


Minor shortcomings:
- The mathematical notation in Eq 1 should be clarified more, e.g in the sentence following it.
- Visualizations of both interferometry and coherence measures would be helpful to help with the intuition of their importance; this could be especially useful since the audience at NeurIPS is not familiar with the technicalities of satellite sensor design.

Questions:
- What is the reasoning for using geographic banding as a form of geographic splits? Why not checkerboard or spatially buffered random splits? I would like to learn about the decision process here.

**Relation To Prior Work:**

Related work is discussed in sufficient detail.

**Summary And Contributions:**

This paper introduces the M3LEO dataset, an Earth observation dataset combining radar & optical data with ground labels (e.g. land cover) and derived products (e.g. elevation maps). The dataset is extensive (>17TB) and focuses on SAR (radar) data, providing interferometry and coherence measurements. The authors package the dataset into a ML-ready Pytorch Lightning interface with high-performance dataloading functionality and functions to include custom data. Lastly, the authors provide baseline evaluations testing the importance of the different bands (mostly optical vs radar) for the included predictive. downstream tasks (e.g. land cover, above ground biomass) and discuss their findings and insights inferred from them.

---

> ### Author Rebuttal · Authors · 2024-08-17
>
> Dear Reviewer,
>
> We thank you for your time and valuable feedback, and have addressed some of your concerns in the top level comment. More specific concerns are addressed below.
>
> *This dataset could be very interesting to evaluate pretrained EO foundation models and assess spectral scaling laws - performance changes across different bands used. Including functionalities to either fine-tune or use embeddings of latest generation EO foundation models such as Prithvi or Clay into the repository would make this a very valuable resource for the ML + EO communities.*
>
> We agree that this is an interesting idea. We have previously pre-trained and fine-tuned foundation model architectures like CLIP, DINO, and MAE (published in arxiv.org/abs/2310.00826, arxiv.org/abs/2310.02048, arxiv.org/abs/2310.00119, arxiv.org/abs/2310.03513), and the corresponding code is in our M3LEO repository on GitHub. We will be providing checkpoints for MAE-based models pretrained on the S1 2020 dataset in light of these reviews. We would be interested to integrate other existing foundation models like Clay or Prithvi into our repository, but would consider this outside the scope of this data paper.
>
> *The clear major shortcoming for me is that, while the dataset is great, it should have been tested with more existing and publicly accessible downstream models. While the authors train their own UNet model to test performance of different bands, no existing pretrained EO foundation model is tested. This is quite a low-hanging effort considering the broad availability of EO foundation models that can process both optical and radar data such as Prithvi and Clay.*
>
> We don’t compare to additional models as we don’t intend for the dataset to serve as a strict benchmarking tool. To produce a fair benchmarking tool, we would have to provide stricter comparisons on which components of the data to use and how to use them. When considering factors such as the different polarisations and orbital directions of SAR data, this would require a very detailed treatment that is beyond the scope of this paper. Please see the top level comment for more detail on how we propose adjusting the paper to make this clearer.
>
> *Visualizations of both interferometry and coherence measures would be helpful to help with the intuition of their importance; this could be especially useful since the audience at NeurIPS is not familiar with the technicalities of satellite sensor design.*
>
> We would be happy to provide these under section 3. Interferometric data often look quite interesting.
>
> *What is the reasoning for using geographic banding as a form of geographic splits? Why not checkerboard or spatially buffered random splits? I would like to learn about the decision process here.*
>
> We decided to split our data based on geographic bands to minimize data leakage and ensure comparable distributions between our training, validation and test set. This data was originally created to explore ideas surrounding geographic generalisability (see https://arxiv.org/abs/2310.00826, https://arxiv.org/abs/2310.02048 on the same data). A totally random split or checkerboarding would lead to too many test tiles being adjacent to training tiles. In addition, contiguous blocks at a continent scale would likely show too much distribution shift - the terrain is very different, for example, at the very western vs eastern edges of Europe. A spatially buffered random split would likely have been acceptable.
>
> We are keen not to lock users into a particular split. We provide the ability to split the data in different ways using GEETiles (github.com/rramosp/geetiles - see “Other usages” in the readme), and changing the split requires only the creation of a new csv file, without any data reslicing.

---

> > ### Comment · Reviewer_eCWi · 2024-08-26
> > **Thanks for the reply**
> >
> > No further comment from my side. I think this is a good paper and I have read the other (more critical) reviews but cannot find anything in them that would convince me towards a lower score.

---

### Official Review · Reviewer_B6VH · 2024-07-23

**Rating:** 5
**Confidence:** 4
**Correctness:** Yes
**Clarity:** Yes

**Review:**

Compared with existing multimodal datasets for Earth observation, the coherence data is the main advantage of the proposed M3LEO dataset. The paper is overall easy to follow. I have the following comments regarding the dataset and the benchmarking results.

Regarding the dataset, there are several limitations:
1. The coverage of the constructed dataset is not global. In this case, the evaluation could be biased.
2. The labels' accuracy is not provided and well-analyzed. For example, there may be errors in the world cover project.
3. The advantages of using the SAR coherence data are not validated in the experiments.

Regarding the benchmarking results:
1. It is not a new finding that combining Sentinel-2 data with Sentinel-1 data can improve performance. This has been widely validated in existing papers.
2. Visualization results should be provided to intuitively analyze the effectiveness of combining multiple data sources.

**Strengths:**

Coregister the proposed dataset to multiple labels and providing the easy-to-use codebase are solid contributions to the community.
Also, constructing the ML-ready dataset including SAR coherence is useful for many remote sensing applications.

**Additional Feedback:**

It would be better If more benchmarking results could be provided for the proposed dataset.

**Documentation:**

The dataset is not uploaded.

**Ethics:**

There are no ethics-related issues.

**Opportunities For Improvement:**

1. Since the coverage is not global, it would be nice if the geographic bias could be analyzed. For example, using other datasets as downstream tasks to evaluate the influence of the dataset coverage.

2. Provide more visualization results to analyze in which cases combining the coherence data would be helpful to improve the performance.

3. A clear comparison with existing datasets should be provided, like the SATLAS [1] dataset, and the SSL4EO-S12 [2] dataset.

[1] Bastani, Favyen, et al. "Satlaspretrain: A large-scale dataset for remote sensing image understanding." Proceedings of the IEEE/CVF International Conference on Computer Vision. 2023.
[2] Wang, Yi, et al. "SSL4EO-S12: A large-scale multimodal, multitemporal dataset for self-supervised learning in Earth observation [Software and Data Sets]." IEEE Geoscience and Remote Sensing Magazine 11.3 (2023): 98-106.

**Relation To Prior Work:**

No

**Summary And Contributions:**

This work introduces M3LEO, a multi-modal, multi-label Earth observation dataset that includes polarimetric, interferometric, and coherence SAR data derived from Sentinel-1, alongside Sentinel-2 RGB imagery. The authors also provide a suite of labeled tasks for
model evaluation. Code and tools to use the data are also complemented with this dataset. The experiments show that SAR imagery contains information that is additional to that extracted from RGB data alone.

---

> ### Author Rebuttal · Authors · 2024-08-17
>
> Dear Reviewer,
>
> We thank you for your time and valuable feedback. Many of your concerns are addressed in the top comment. Specific concerns are addressed below.
>
> *2. The labels' accuracy is not provided and well-analyzed. For example, there may be errors in the world cover project.*
>
> This is commonly overlooked in Earth Observation and a good idea to highlight. We are happy to add additional information/references on independent validation of the original satellite products (e.g. as seen in worldcover2020.esa.int/data/docs/WorldCover_PVR_V1.1.pdf), but this is an issue in general with EO products, rather than with our work particularly.
>
> *3. A clear comparison with existing datasets should be provided, like the SATLAS [1] dataset, and the SSL4EO-S12 [2] dataset.*
>
> We will include a clearer comparison to existing datasets within the text.

---

### Official Review · Reviewer_XVwA · 2024-07-25
**More work is needed**

**Rating:** 4
**Confidence:** 5
**Clarity:** The paper is well written.

**Review:**

The paper is mostly well written with some informative parts regarding the SAR domain. The dataset is unique and promotes the usage of SAR data by ML practitioners.

The related work section is particularly poor, omitting many works investigating DL methods for SAR data as well as many SAR datasets, including ones that contain Interferometric SAR. Some examples of such datasets would be:
Sen1Floods11, WorldFloods, CAU-Flood, UrbanSARFloods, S1SLC_CVDL, Hephaestus and many more.

Even more, the argument at L97 that previous works attempt to create foundation models with SAR data at small geographic scales is not correct. Many works use global SAR data while many global datasets exist that combine Sentinel-1 SAR data with optical data e.g. SSL4EO-S12 and Sen12MS.

The experimental section is rather weak, using only a small UNet to demonstrate the quality of the datasets. Even more, the experiments omitted the chips coming from Europe and randomly selected a subset of the rest of the data (i.e 10% of the training and validation sets) for the experiments. With no elaborate experimental section or experiments with multiple seeds it is very difficult to draw conclusions about the dataset, and as a result the Discussions section gets weakened too.

The dataset is interesting despite its shortcomings but the motivation behind this work is not clear at all. ML3LEO is a huge dataset spanning a very short period of time and selected areas, with aligned SAR products paired with maps that could potentially serve as ground truth labels. It is not clear how these products will work in synergy (and why one would need it) and it is not demonstrated in the experimental section either. It seems that the authors leave it to the reader to decide how this dataset is going to be used making its foundations rather weak. There is no clear motivation for the choice of the downstream tasks. Similarly there is no motivation behind the choice of all these SAR products to solve the selected downstream tasks.  ML3LEO has the potential to be a valuable dataset for the community but I feel that it is not ready yet.

**Strengths:**

M3LEO provides an ML ready source of GRD, Interferometric and Coherence SAR products at a large scale. This is important with not many (if any) similar sources of SAR data. This data is paired with land cover maps and DEMs offering some downstream applications to the input data.

**Additional Feedback:**

n/a

**Correctness:**

The current paper does not thoroughly investigate the value of the dataset. While a set of downstream tasks is used to evaluate the data quality and extract insights, the limited number of experiments prevents drawing valid conclusions. Additionally, other methods for exploring dataset properties—such as visualizations, estimating dataset diversity, comparing with other datasets, and investigating distribution shifts—could provide more meaningful insights than basic supervised learning experiments.

**Documentation:**

There is detail on data collection even though the temporal resolution of the data is not mentioned (e.g. we acquire SAR data every X days). The work seems well engineered.

**Ethics:**

No ethics concerns.

**Limitations:**

The authors discuss the limited spatial and temporal coverage and justified their choice with a promise about future extension of the dataset. Of course, for the scope of this review, I have to assess the current state of the proposed paper and dataset. Even more, they discussed the lack of visualizations as well as the limited benchmarking, while encouraging the readers/domain experts to proceed with their own investigations.

**Opportunities For Improvement:**

Temporal coverage: To effectively make use of such data, the dataset needs to have a reasonable temporal coverage e.g from 2015  to 2023. Many Remote Sensing applications rely on the study of the temporal evolution as well as the changes on the Earth’s surface.

Spatial coverage: The small spatial coverage is not justified. Indeed the size of the dataset is large but a smarter scene sampling can help mitigate this problem. Also no data from Australia and Africa are provided (among others).

Motivation of this work: The motivation behind the introduction of M3LEO (even though it is an interesting dataset) is not clear at all. A better positioning can help the reader understand the value of this work. This applies to the choice of the land cover maps. It feels like the authors plugged in any kind of ground truth just to assess the datasets.  M3LEO could be potentially utilized for large-scale self-supervised learning but it would be a bit contradictory given the short temporal coverage.

Synergy of the SAR products: Are the provided SAR products intended to be utilized together? What is the motivation behind this alignment? Would the dataset benefit from providing the unaligned data too? The paper should provide such explorations to the reader and guide the community towards a direction that could make the most of this dataset.

**Relation To Prior Work:**

The related work needs major improvements. It currently falls short in depicting the state of the art and as a result fails to motivate the current work and showcase its importance. The dataset however seems unique.

**Summary And Contributions:**

The authors introduce a new dataset consisting of 3 Sentinel-1 SAR products acquired from public data providers. The products are a) SAR GRD, b) Interferometric SAR and c) coherence. The total volume of the dataset is 17.5TB with 10M chips. The dataset has a very limited temporal coverage with data spanning only 2020 and a limited spatial coverage i.e it covers Europe, China, Middle East, a small proportion India, Pakistan, United States, and South America. The temporal resolution of the data is not mentioned explicitly. Additionally, the authors provide the respective Sentinel-2 RGB data, omitting to include the rest of the spectral bands of Sentinel-2 with no clear motivation.  The experimental section is rather weak, while the motivation of this work is not clear.

---

> ### Author Rebuttal · Authors · 2024-08-17
>
> Dear Reviewer,
>
> We thank you for your time and valuable feedback. Many of your concerns have been addressed in the top comment, as they are relevant to other reviewers also. Some specific concerns are addressed below.
>
> *The temporal resolution of the data is not mentioned explicitly/temporal resolution of the data is not mentioned (e.g. we acquire SAR data every X days)*
>
> SAR amplitude data is currently provided as seasonal averages. For S1, it’s not possible to make a blanket statement of X days on revisit times, as they depend on location and the answer is different depending on whether a pass in a different direction is considered as a repeat or not. The maximum amount of time between having repeated acquisitions for both directions at any location is at worst 12 days, but it could be as low as 6. The ascending/descending acquisitions will not occur on the same day. Coherence and interferometric data require the selection of date-pairs rather than individual dates - we outline the selections we made for date-pairs in the text.
>
> *The related work section is particularly poor, omitting many works investigating DL methods for SAR data as well as many SAR datasets, including ones that contain Interferometric SAR. Some examples of such datasets would be: Sen1Floods11, WorldFloods, CAU-Flood, UrbanSARFloods, S1SLC_CVDL, Hephaestus and many more.*
>
> We thank the reviewer for these citations and would be happy to include them. We clarify that our data is unique in its combination of size, geographic coverage and coregistration with numerous satellite products.
>
> *Even more, the argument at L97 that previous works attempt to create foundation models with SAR data at small geographic scales is not correct. Many works use global SAR data while many global datasets exist that combine Sentinel-1 SAR data with optical data e.g. SSL4EO-S12 and Sen12MS.*
>
> We thank the reviewer for pointing this out - the comment about small geographic scales should have been removed previously. We believe the point regarding optical data on L99 still stands. L100-101 applies to work using SAR without optical data.
>
> *It seems that the authors leave it to the reader to decide how this dataset is going to be used making its foundations rather weak.*
>
> We intentionally avoid constraining users by prescribing specific ways to apply the dataset, and encourage more creative explorations. To support this, we are happy to direct users to examples of how the data has been used in past research (see arxiv.org/abs/2310.00826, arxiv.org/abs/2310.02048, arxiv.org/abs/2310.00119, arxiv.org/abs/2310.03513), along with the use cases outlined in Section 1.
> We believe that specifying a rigid evaluation protocol could stifle the development of new approaches, as it may lead to work overly tailored to our specific testing methodology.
> *Spatial coverage: The small spatial coverage is not justified. Indeed the size of the dataset is large but a smarter scene sampling can help mitigate this problem. Also no data from Australia and Africa are provided (among others).*
>
> We refer to the top level comment regarding further justification on the spatial coverage. It is not immediately clear what constitutes a smart sampling scheme when selecting from complex-valued, actively illuminated radar data vs. optical data and we would not wish to bias results by pre-sampling the data for users. We are actively researching sampling schemes and hope that this data will enable others to do the same.
>
> *M3LEO could be potentially utilized for large-scale self supervised learning but it would be a bit contradictory given the short temporal coverage.*
>
> It is possible to draw insights regarding geographic generalisability in self-supervised learning from the single-year that is currently in the public-facing repository (https://arxiv.org/abs/2310.02048, https://arxiv.org/abs/2310.00826), but refer to our top level comment regarding additional years.
>
> *Synergy of the SAR products: Are the provided SAR products intended to be utilized together? What is the motivation behind this alignment? Would the dataset benefit from providing the unaligned data too?*
>
> We do not wish to box in users regarding how they choose to use the data downstream. We provide a number of example use cases for interferometric SAR within the introduction and would be happy to signpost readers to previous work using this dataset specifically (e.g. https://arxiv.org/abs/2310.02048, https://arxiv.org/abs/2310.00826). The unaligned data is hosted on various other platforms and although it requires some preprocessing for use, we do not feel that re-hosting it all in the same place would be beneficial enough to justify the storage requirements.

---

> > ### Comment · Reviewer_XVwA · 2024-08-31
> > **Thank you for the rebuttal**
> >
> > I would like to thank the authors for their thorough rebuttal and their sincere effort to address all reviewers' comments, including mine.
> >
> > I find myself in agreement with many of the points raised by Reviewer jNsP, particularly regarding both the strengths and weaknesses of this work, though our final ratings differ. As I noted in my initial review, the paper is well-written, and the dataset is excellently engineered. The key question, therefore, is whether M3LEO, with its specific design choices and potential applications, represents a valuable contribution to our community.
> >
> > Upon further consideration, and partly influenced by Reviewer jNsP's perspective, I believe that despite its current limitations, M3LEO—and its future iterations—holds promise for advancing the investigation of the SAR modality in ML4EO applications.
> >
> > Consequently, I am **updating my score to 6**, with the expectation that the authors will fulfill their commitments outlined in the rebuttal (particularly the main one), including the revisions to the related work section, for the camera-ready manuscript.

---

### Official Review · Reviewer_jNsP · 2024-07-25
**M3LEO Review**

**Rating:** 7
**Confidence:** 4

**Review:**

Large ML datasets for EO, especially Sentinel, are relatively common. Although the vast majority of the literature focuses on Sentinel-2 (optical imagery) with Sentinel-1 (SAR) included as an afterthought, this paper takes the exact opposite approach. A dataset of this size including polarimetric, interferometric, and coherence SAR data is not only incredibly novel, but also incredibly impactful. Although I have a few complaints about the handling of Sentinel-2 or sampling strategy (see "Opportunities for Improvement"), I have an overall positive impression of the potential of this dataset for SAR foundation model pre-training.

Pros:

* One of the largest (if not the largest) ML-ready multi-modal SAR datasets in existence
* Inclusion of just about every possible SAR datatype someone might need to work with
* Careful attention to detail and handling of SAR data, which requires domain expertise to work with
* Multiple software libraries for reproducing or contributing to this dataset in the future

Cons:

* Sentinel-2 data is RGB-only and smoothed, making it practically useless for most applications
* Dataset is almost global, but ignores entire continents (Africa) and important biomes (polar, rainforest), exacerbating already existing data biases
* No benchmark comparisons of this dataset with existing Sentinel-1/2 datasets (e.g., SSL4EO-S12)

**Strengths:**

The M3LEO dataset fills an important gap in existing datasets, which tend to include primarily Sentinel-2 MSI and Sentinel-1 GRD data. This is one of the first ML-ready datasets to include SAR interferometry and coherence data. The authors include an extremely detailed background on SAR that greatly helped this reviewer understand the complexity of the data.

The multi-modal and multi-task nature of the dataset enables the development of exciting future foundation models, which have the potential to have an impact across a large number of downstream domains from natural and manmade disaster monitoring to biomass estimation.

The dataset is enormous, with 10M data chips (17.5 TB) covering 14.1% of Earth's land surface. The authors also publish code for integration with PyTorch Lightning (M3LEO) and GEE (geetiles, sartiles), allowing the dataset to be further expanded in the future.

P.S. Thank you for defining "tile" and "chip", as these have different definitions in almost every paper I read!

**Additional Feedback:**

* Adjacent citations can be combined, so \cite{foo}\cite{bar}\cite{baz} becomes \cite{foo,bar,baz} and [1][2][3] becomes [1–3]
* Lines 42, 45, 56, 90, 101, 108, 173, etc: an em dash "---" should be used instead of a hyphen "-" for asides
* Line 52: also worth mentioning that InSAR is useful for monitoring the growth of magma chambers underneath volcanoes
* Lines 59, 121: also worth clarifying that amplitude and phase are often stored and represented using complex numbers, which are difficult for most computer vision pipelines (I/O, transforms, and models)
* Line 115: met -> met with
* Line 77: if you're looking for additional SSL foundation models to cite, https://arxiv.org/abs/2405.04285 has a pretty good list
* Section 3: there should be a non-breaking space between numbers and their units (e.g., 10 m/px, not 10m/px)
* Lines 167, etc: "[74] and [75]" should be \citet, not \citep since they are words: https://www.ece.ucdavis.edu/~jowens/commonerrors.html
* Section 3.5: units should be added to image dimensions (presumably pixels, but could also be meters)

**Clarity:**

* Line 53: first sentence feels out of place, but important, so should probably move into the middle of the paragraph
* Line 91: 14.1% of the *land* surface of the Earth
* Line 92: last sentence feels out of place, but important, so should probably moved to a later paragraph
* Line 177: RGB is the only product for which the resolution is not specified
* Line 177: surface reflectance or top of atmosphere?
* Line 237: the authors introduce code for SSL, but SSL is not used in this paper?
* Section 3: this should be split into multiple sections, only 3.1–3.3 relate to datasets, 3.4–3.6 should likely have their own section
* Line 281: ViTs can be used as encoder backbones in U-Nets, ViTs and U-Nets are not mutually exclusive
* Table 2: thank you for adding units to RMSE!

**Correctness:**

While this dataset focuses heavily on Sentinel-1 SAR data, the majority of labeled tasks are not well suited for SAR data. Specifically, biomass and vegetation cover seem to be common uses of SAR, but land cover, built surface, and elevation (not change in elevation) are not. Some justification as to why these specific datasets were chosen or why they might be useful for model training would be helpful.

This dataset lacks a sampling strategy, as every possible image location is included in the dataset. Although this results in a very large dataset, this does not necessarily improve dataset diversity. Approximately two thirds of the Earth's land surface are desert or forest, resulting in less image diversity than other sampling strategies such as [SeCo](https://arxiv.org/abs/2103.16607). In my own (admittedly limited) experience, models pre-trained on [SSL4EO-S12](https://arxiv.org/abs/2211.07044) outperform models pre-trained on [SatlasPretrain](https://arxiv.org/abs/2211.15660) despite Satlas being newer and larger, likely due to less dataset diversity.

The authors made the difficult but correct choice to avoid common data augmentations such as rotation or flipping due to the inherent directionality and polarization of SAR data. All experiments are conducted in a sound way.

Line 331: "To our knowledge, this is the only work to date including SAR datatypes beyond polarimetry in an ML-readable format."
This may be a stretch. A quick Google search turned up many InSAR datasets designed for ML: https://arxiv.org/abs/2204.09435, https://dx.doi.org/10.21227/dhxt-5g91, https://dx.doi.org/10.35097/1700. I don't know if these are all _easily_ ML-readable, but that's relatively subjective. Maybe it would be safer to claim that this is the largest multimodal SAR dataset ever created?

**Documentation:**

Data collection and curation is well-documented. The inclusion of an extensive datasheet made available in the supplementary materials greatly improves the documentation on data provenance and maintenance.

The evaluation procedure is also well documented, and code is made publicly available.

https://github.com/spaceml-org/M3LEO is missing a license, meaning that no one is given permission to use, modify, or redistribute the software for any purpose. Please add a license to this repo. Although the supplementary materials suggest that the code may be intended for release under CC-BY-SA-4.0, I would recommend MIT/Apache instead, as CC licenses are not intended for source code.

While on the topic of software, it is worth noting that M3LEO, geetiles, and sartiles all lack unit testing or CI. This makes the reproducibility of the experiments on other platforms or future versions of software questionable. However, this is not a software track paper, so I tried not to let this influence my review score. I would love to see unit tests added in the future to improve these software contributions. You could even consider contributing some of the data loaders and PyTorch Lightning code to a library like [TorchGeo](https://github.com/microsoft/torchgeo) to increase the availability of the dataset to a wider audience. But again, this is simply a suggestion, and does not influence my review score.

**Ethics:**

I do not see any obvious ethical concerns with this paper. I would have to confirm the individual licenses, but all data used in the creation of this dataset comes from ESA/NASA/USGS and tends to be released under the public domain. The dataset does not contain human subjects, and the labeled tasks are relatively innocuous (and already public). The only possible concern is whether or not the dataset is representative of all global regions, in particular Africa, which is historically excluded from most datasets due to lack of data.

**Limitations:**

The authors are very careful about adding qualifiers to their findings:

* "we do not claim conclusively that either S1GRD or S2RGB achieves greater performance for any particular task."
* "so cannot confirm whether including additional S2RGB bands such as Short-Wave Infrared (SWIR) would reduce this effect."

The authors also provide many great suggestions for ways to further improve the utility of their dataset, some of which I comment on above in "Opportunities for Improvement". My only suggestion would be to discuss some of my comments below in "Correctness" to address the possible limitations of the labeled datasets and data collection process.

The authors do not explicitly discuss possible negative societal impacts of their work, but I don't see any off the top of my head. All data collected here is already publicly available, it is simply repackaged into a more ML-accessible format.

**Opportunities For Improvement:**

The most obvious room for improvement is in the Sentinel-2 imagery included in this dataset, which feels more like an afterthought than a useful contribution. The fact that only RGB data is provided severely hampers the usefulness of this multispectral image source. Many papers have found that NIR and SWIR data is significantly more important than the visible spectrum for a number of applications, especially relating to vegetation/agriculture and land cover mapping. The authors do not explain why these additional spectral bands were excluded (to reduce the total storage volume?), but I don't know if I've ever seen a single paper use RGB-only Sentinel-2 data.

On a related note, the Sentinel-2 data is computed as means per-month, resulting in smoothing of extremes like sparse cloud cover. Without confirming this myself, I am dubious of whether a model trained on such smoothed data would be able to transfer to unsmoothed, raw Sentinel-2 imagery. This limits the usefulness of the dataset for SSL pre-training.

Speaking of SSL pre-training, the vast size of this dataset makes it very attractive for foundation model pre-training. Given that the authors are already working on SSL in their PyTorch Lightning codebase, I'm guessing this is already planned for future work. As much as I would love to see new foundation models proven to outperform existing ones, I think this paper already contains more than enough contributions to see it published. I would instead suggest focusing on the other improvements and saving foundation models for a future paper.

As mentioned in more detail in "Correctness" below, the inclusion of different dataset labels more specific to SAR data would improve the usefulness of the dataset labels. But the images alone are already extremely useful for SSL.

I am sad to see Africa left out of yet another almost-global dataset. I realize this was done due to label constraints, but I'm skeptical of the usefulness of the labels for SAR anyway. It is also worth noting that this dataset lacks imagery from polar regions (Arctic, Antarctic) and rainforests (Amazon, SE Asia). However, this can also be said about SeCo and SSL4EO-S12, so I don't see this as a reason to reject the paper, just a possible way to improve it.

This is a minor comment, but I don't see a single figure/table until page 6. If you can find the room, I think this paper would benefit from example dataset visualizations or at least some kind of summary figure earlier in the paper.

**Relation To Prior Work:**

The authors provide a detailed history of deep learning and EO, EO foundation models, EO applications, and EO challenges. Honestly, this might even be too broad. Of greater importance is the connection to SAR data (why it can be important but difficult to work with), which the authors do a great job discussing. The paper does not explicitly describe what M3LEO has that previous datasets don't, but implicitly explains this by discussing the limitations of existing datasets. For example, SSL4EO-S12 is also multi-modal (Sentinel 1 + 2) but only has a single SAR datatype, lacks any labels, and is considerably smaller.

**Summary And Contributions:**

This paper introduces a new multi-modal, multi-label Earth observation (EO) dataset (M3LEO) with specific emphasis on the wide variety of synthetic aperture radar (SAR) data products available with Sentinel-1. This is one of the largest EO datasets I've ever seen with 17.5 TB of data and 10M data chips. In addition to the large dataset, the authors also present not one but three software libraries created for the purposes of this work. This includes tools for data downloading, preprocessing, and tiling (geetiles, sartiles) and experiment evaluation (M3LEO). Integration with Google Earth Engine (GEE) and PyTorch Lightning makes these tools both powerful and user friendly.

Specific contributions of the paper:

* The M3LEO dataset, a large-scale, multi-modal, multi-label EO dataset composed of data from Sentinel-1 (polarimetric, interferometric, and coherence SAR), Sentinel-2 (RGB), and several global semantic segmentation tasks
* The M3LEO code, including data loaders and PyTorch Lightning trainers for experiment evaluation
* The geetiles and sartiles libraries, for data downloading from GEE and data preprocessing
* Further experimental evidence to suggest that SAR data can compliment existing optical imagery models

---

> ### Author Rebuttal · Authors · 2024-08-17
>
> Dear Reviewer,
>
> We thank you for your time and valuable feedback, and have addressed your specific concerns below.
>
> *Sentinel-2 data is RGB-only and smoothed, making it practically useless for most applications*
>
> Regarding the inclusion of only the RGB bands - we made this decision at the time to reduce storage requirements as our main focus was specifically on SAR data with Sentinel-2 being added later. However, we have received this feedback several times both within and outside of these reviews and we will be adding further bands to the Sentinel-2 data (SWIR, NIR) - but we can’t provide a firm timeline on this. Regarding the smoothed data - we have found that the averaged data is visibly reasonable and performs reasonably on a variety of tasks, but we will investigate including cloud-free mosaics composed of single acquisitions at the same time as adjusting the bands.
>
> *No benchmark comparisons of this dataset with existing Sentinel-1/2 datasets (e.g., SSL4EO-S12)*
>
> We are hesitant to provide comparisons regarding e.g. model performance using M3LEO vs other datasets to avoid confounding results with differences due to e.g. model choice or pretraining schemes. We would be happy to provide a more explicit comparison regarding the content of the datasets, however.
>
> *resulting in smoothing of extremes like sparse cloud cover*
>
> Although the data is smoothed, only cloud-free pixels are included in the average. We will make this explicit in the text.
>
> *While this dataset focuses heavily on Sentinel-1 SAR data, the majority of labeled tasks are not well suited for SAR data…*
>
> Although we do not claim that SAR is the best data type for these tasks, we believe they are still a good litmus test of the data behaving as we expect it to. We have found in other work on the same data (arxiv.org/abs/2310.00826) that performance on these (or very similar) tasks seems to show high responsiveness to self-supervised learning schemes. It is important not to draw too many conclusions from previous work applying DL to these tasks, as it often operates on amplitude-only SAR data without consideration of phase/interferometric information (not provided via Google Earth Engine).
>
> *This dataset lacks a sampling strategy, as every possible image location is included in the dataset. Although this results in a very large dataset, this does not necessarily improve dataset diversity…*
>
> We agree that the use of a sensible sampling strategy is extremely important. We are actively investigating this in separate pieces of work, but a good sampling strategy for actively-illuminated, radar images with phase information is not obvious. These images can look very different to optical imagery. Certain types of vegetation, for example, are totally invisible in this data. Providing the full data enables others to perform research on these sampling strategies, whereas other datasets force use of a subset of data generated with a pre-defined strategy.
>
> *Line 331: "To our knowledge, this is the only work to date including SAR datatypes beyond polarimetry in an ML-readable format." This may be a stretch…*
>
> We are grateful for these additional citations - we believe our dataset is unique in its combination of size, geographic scope and coregistration with additional data. We will clarify this in the text.
>
> **Regarding additional comments on specific lines with small mistakes or unclear wording** - we would like to add that we are extremely grateful for the reviewer lending us an extra set of eyes, and we will fix or clarify all of these points.

---

> > ### Comment · Reviewer_jNsP · 2024-08-27
> > **Response to Rebuttal**
> >
> > Thanks for your rebuttal. I agree with most of the points you make, and your desire not to overly constrain the dataset for a particular application. I agree that the SAR data is the primary contribution of this paper, and the RGB imagery and labels are more afterthoughts. It would be interesting to include EW swaths and HH/HV polarizations in a future expanded version, but I still think the current version of this dataset is sufficient for publication. I will stand by my previous review score (7, accept) and wish you luck with the other reviewers.
> >
> > If by some chance this paper does not get accepted, I hope my comments were helpful in deciding what else is worth exploring. Honestly, a paper only on SAR images for pre-training purposes and some simple benchmark results on existing datasets would be fine with me. Including all bands of S2 would also be worth adding if you have time. But I would downplay the collected labels and instead focus on improving the SAR data as much as possible. That really is what makes this paper unique and special.

---

### Author Rebuttal · Authors · 2024-08-17

Dear Reviewers,

There were a few common themes present in several reviews. We propose relevant changes in this comment rather than repeating them in response to individual reviews.

In addition to the open-access miniset on HuggingFace, **we have provided instructions to access a private data repository in an official comment to all reviewers via OpenReview**, so that the reviewers can verify the completeness of datasets during the review process. We were unable to provide this prior to the rebuttal stage without announcing this link openly.

If the reviewers feel that we have addressed their concerns, we kindly ask for a score update.

1. Reviewers felt that the spatial coverage was poorly justified in the text. Although we do mention that we limited all datasets to the coverage of the interferometric data, we are happy to add extended reasoning regarding why the coverage looks like this to the text.
We refer reviewers to Figure 21 at https://sentiwiki.copernicus.eu/web/s1-mission for coverage map and point to the following:


   a. The acquisition mode of the satellite is different in polar regions vs. most of the earth’s land surface, using Extra Wide (EW) swath vs Interferometric Wide (IW) swath. The surface is illuminated/measured differently in these regions e.g. the azimuth steering angle of the radar emitter is 0.6 degrees for IW vs 0.8 for EW The data we provide is currently IW exclusively

   b. Different polarisations are used in the polar regions (HH, HV rather than VV, VH).

   c. Much area of the Earth is covered only by one direction of satellite travel (e.g. the Brazilian amazon is mostly descending only)

The data that we include covers a large portion of the area in which the Sentinel-1 constellation operated with IW acquisition mode,   dual polarization (VV+VH) and both ascending and descending orbits.

We could include other regions - but it is not straightforward to say that this would make our data less biased. While it is true that universal coverage would reduce geographic bias, we would introduce a new systematic bias in how the data was measured, depending on location. It is not clear how well models would transfer to regions with different polarisations, or how to handle the systematic absence of certain polarisations in different regions, for example. We believe that these issues deserve a complete and detailed treatment that is beyond the scope of what we could include in a single piece of work.

2. Reviewers mentioned that the experiments were not particularly insightful. We are keen to emphasize that we do not intend for this dataset to be a benchmarking framework, but we acknowledge that the text may give this impression. We propose moving the experimental section to the appendices, as it is not crucial to understanding the dataset, and adding an extended section on data visualizations and figures analyzing distribution shifts between train/test splits, geographic regions etc. to the main text. We show an example of such a figure that was to hand in the PDF attached. We would be able to add more in-depth comparisons, such as comparing distribution shifts, during revisions. We feel this does not change the main value proposition or message of the work. It does not require training models, so is achievable by the camera-ready deadline.

3. Reviewers felt that Sentinel-2 was not useful without additional bands. We would like to emphasize that our major contribution is the provision of SAR data (including interferometric data), and that the dataset is still very valuable with full Sentinel-2 inclusion. We agree that these bands would be valuable, and we will be adding SWIR/NIR bands to Sentinel-2 but we cannot make a promise about when this would be.

4. It was mentioned that that multitemporal data would be preferable. Sentinel-1 data plus Sentinel 2 (RGB) have been processed for 2018-2021, which reviewers can verify via the access described in the comment. Some tasks are available for multiple years; this is mostly governed by which years the data exists for, rather than the lack of us processing it. See Table 1 in the attached PDF for a breakdown. We had intended to only include the data on which we ran experiments, but we would be happy to upload these to the public repository if we were to provide analysis on distribution shifts instead (per comment 2). As the data has already been processed, the cost of doing this is not high - they would simply need to be uploaded. We would not be able to run experiments involving model training on these data.

5. Justification for choice of tasks. We focused on selecting downstream tasks that are broadly desirable when using EO data, rather than choosing tasks that might favor SAR data for the sake of reporting positive outcomes. We do not think it is guaranteed that previous findings will hold when considering SAR with interferometric information. However, we emphasize that this work is not intended to form a strict benchmarking framework. Such datasets (Land cover, AGB, Built surface) provide useful information about surface properties that could be used in other ways, such as in sampling schemes. We will provide this framing in the text with the changes outlined in comment 2. Finally, additional “task” datasets (Flood events, burned area detection) are available at the private repository and would be uploaded, but we cannot run experiments on them. SAR data may be more suited to these tasks.

6. Self-supervised learning/models. Some comments were made on the inclusion of code for SSL models despite them not being in experiments. We would signpost readers to previous work using this data (arxiv.org/abs/2310.00826, arxiv.org/abs/2310.02048, arxiv.org/abs/2310.00119, arxiv.org/abs/2310.03513) more explicitly in the text. We have checkpoints for MAE-based models pretrained on the entire 2020 S1 dataset, which can be verified using the access instructions in the official comment.

---

> ### Author Rebuttal · Authors · 2024-08-19
>
> Edit: We have discussed in detail amongst ourselves and are confident the additional Sentinel-2 bands could be put in place prior to the dates of the conference. We would not be able to rerun the experiments.

---

### Author Response · Authors · 2024-08-17
**Access Instructions**

Datasets: https://console.cloud.google.com/storage/browser/fdl2023-sar-datasets
Checkpoints: https://huggingface.co/M3LEO

Checkpoints for MAE (ViT Base) pretrained on Sentinel-1 data for all regions (2020) will be uploaded to HuggingFace on or prior to Mon 19th Aug. Regional checkpoints for additional models (UNet, MAE, DINO, CLIP) will be uploaded at a later date.

Individual datasets can be seen under folders such as fdl2023-sar-datasets/conus/conus_partitions_aschips_293d95e3ee589 (for contiguous United States)

**Downloading on GCP is locked to requester pays**, as we cannot verify the source/user of the request if we were to enable open download, which carries a monetary cost to us per-file downloaded. You should still be able to browse the folders. **Please take care not to accidentally download data should your google accounts be linked to a Google Cloud Platform Project, as your accounts will be charged.**

---

### Decision · Program_Chairs · 2024-09-26

**Decision:**

Accept (Poster)

**Comment:**

General info for the program chair:

For this paper, Reviewer XVwA mentioned that he/she would upgrade the score to 6, but didn’t change it in the system. Should we take this into account, the paper should have a final average score of 6.5, instead of 6.

Dear Authors,

I have reviewed the comments from the reviewers and your response.

The M3LEO dataset, especially its focus on Sentinel-1 SAR data (including interferometric and coherence data), was seen as a major contribution. Several reviewers praised the scale and potential impact of this dataset for Earth observation (EO). The codebase and data tools provided, including integration with PyTorch Lightning and Google Earth Engine, were also appreciated for their ease of use and potential for extending the dataset. In addition, the dataset's large scale and its potential to be used for SSL and foundation models were seen as significant.

However, the paper still has potential to be further improved, among others:

-	Spatial and temporal coverage: Several reviewers expressed concerns about the limited geographic and temporal scope of the dataset. The dataset does not cover some regions, such as Africa, Australia and polar regions, and focuses only on 2020.

-	The exclusion of certain Sentinel-2 spectral bands (such as NIR and SWIR) was considered as a limitation, reducing the usefulness of the dataset for tasks related to vegetation and land cover. This aspect is particularly important for Earth observation domain applications. This is promised to be addressed by the authors in the final paper.

-	The experiments presented were considered underwhelming, with a lack of depth and variety in model testing, e.g. only the reliance on a small UNet model and the lack of testing on more diverse or pre-trained models. I agree with the authors that the purpose of the dataset is not designed to benchmark foundation models (FMs). Nevertheless, it is interesting to evaluate the dataset’s potential on pretraining FMs.

-	A more detailed analysis of the dataset’s biases and label accuracy would be helpful.


I hope these comments and suggestions can help authors to further improve your work.